# Non-Adaptive Adversarial Face Generation

**Sunpill Kim    Seunghun Paik    Chanwoo Hwang    Minsu Kim    Jae Hong Seo**[*]

Department of Mathematics & Research Institute for Natural Sciences, Hanyang University

{ksp0352, whitesoonguh, aa5568, iayaho3248, jaehongseo}@hanyang.ac.kr

## Abstract

Adversarial attacks on face recognition systems (FRSs) pose serious security and privacy threats, especially when these systems are used for identity verification. In this paper, we propose a novel method for generating *adversarial faces*—synthetic facial images that are visually distinct yet recognized as a target identity by the FRS. Unlike iterative optimization-based approaches (e.g., gradient descent or other iterative solvers), our method leverages the structural characteristics of the FRS feature space. We figure out that individuals sharing the same attribute (e.g., gender or race) form an attributed subsphere. By utilizing such subspheres, our method achieves both non-adaptiveness and a remarkably small number of queries. This eliminates the need for relying on transferability and open-source surrogate models, which have been a typical strategy when repeated adaptive queries to commercial FRSs are impossible. Despite requiring only a single non-adaptive query consisting of 100 face images, our method achieves a high success rate of over 93% against AWS's CompareFaces API at its default threshold. Furthermore, unlike many existing attacks that perturb a given image, our method can deliberately produce adversarial faces that impersonate the target identity while exhibiting high-level attributes *chosen by the adversary*.

## 1  Introduction

Computer vision has advanced significantly with the development of Deep Learning (DL) technologies, which enable the extraction of discriminative features from images and have proven useful in various tasks such as classification and recognition. For example, with sufficient data, DL models demonstrate remarkable accuracy in image classification [83, 26, 85] and face recognition [16, 3, 40].

However, the high accuracy of DL models has typically been evaluated using naturally generated (i.e., unaltered) images, and studies have shown that adversarially generated images, called adversarial examples, can fool DL models with high probability [23, 53, 76]. Adversarial examples are artificially generated images that are perceived differently by humans and DL models. Both the generation of adversarial examples and the development of defense and detection methods are active areas of research, as adversarial examples present significant security and privacy risks in the practical deployment of DL [53, 23, 31]. For example, DL-based Face Recognition Systems (FRSs) are widely used for mobiles and website logins, as well as for access control of buildings and airports [38, 44, 14]. In such systems, adversarial examples pose significant security and privacy risks [82, 79]. In this paper, we focus on FRS, but we believe that the techniques we developed can be spread to other biometric fields such as voice [46, 12, 47, 25]. This is a natural extension, as FRS is a representative biometric authentication method, and the core objective of biometric systems is to achieve strong intra-class compactness and inter-class separability—principles that apply across various modalities.

Several methods have been proposed to generate adversarial examples that can fool FRSs, and these are typically categorized based on the adversary's capabilities. If the adversary has full access to the

---

[*]Corresponding author

39th Conference on Neural Information Processing Systems (NeurIPS 2025).

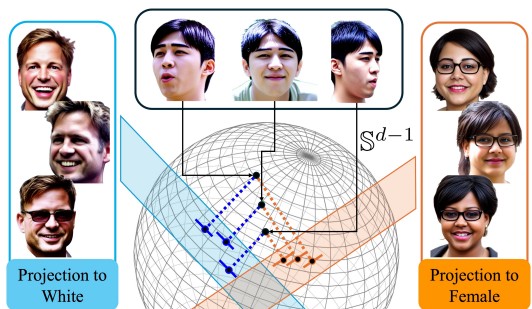

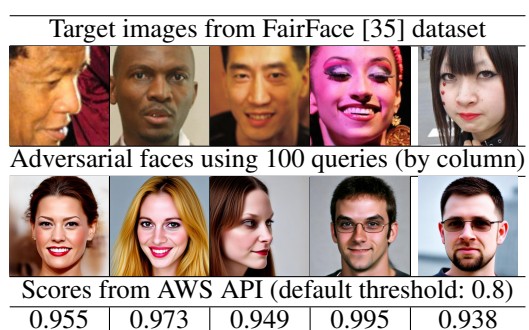

| Target images from FairFace [35] dataset |
| --- |

| Adversarial faces using 100 queries (by column) |
| --- |

| Scores from AWS API (default threshold: 0.8) | | | | |
| --- | --- | --- | --- | --- |
| 0.955 | 0.973 | 0.949 | 0.995 | 0.938 |

Figure 1: The core idea is to project a feature vector $\vec{x} \in \mathbb{S}^{d-1}$ onto an attributed subsphere $\mathbb{S}_f^k$.

Figure 2: Examples of adversarial faces attack successfully against the AWS API [65]

target DL model's parameters, the attack is categorized as a "*white-box attack*" [53, 23, 18, 49, 76]. If the adversary cannot access the model's parameters but can query, it is considered a "*black-box attack*" [31, 7, 4]. Black-box attacks can further be categorized based on the type of query result. For example, if the result is a final prediction, such as an identity label or a true/false verification outcome, it is a "*hard-label/decision-based attack*" [4, 19, 8]. If the result includes logits or similarity scores, it is a "*soft-label/score-based attack*" [7, 31, 24, 71, 50]. Regardless of these categories, "*GAN-based*" approaches have been proposed to improve the imperceptibility [82, 28, 68, 45] of adversarial examples. Ordinary adversarial examples are generated by directly modifying pixel values, which can occasionally introduce unnatural artifacts that are noticeable to humans. To address this limitation, GAN-based methods instead search the latent space of a GAN to synthesize natural-looking images. Other classifications include "*optimization-based*" approaches, which solve discrete and non-continuous problems in the hard-label black-box setting [10], and "*transferable attacks*", which exploit the transferability of adversarial examples across DL models to bypass limited access in the black-box setting [52, 62].

Although the above adversarial attacks fall into different categories, they all *iteratively* solve optimization problems under constraints that ensure the generated examples remain imperceptible to humans. For example, the PGD attack [53] is an iterative process that consists of two steps: (1) finding a perturbed image whose feature vector is far from that of the original, and (2) projecting it onto a set of small perturbations to ensure imperceptibility to humans. GAN-based attacks [82, 28, 68, 45] also involve an iterative optimization process subject to two objectives: maximizing the distance from the original image and minimizing perceptibility by humans. This is because these attacks rely on iterative solvers such as gradient descent [53, 23, 18, 49, 76] and randomized gradient-free methods [7, 71, 10]. However, these iterative solvers are cumbersome, especially in black-box settings, because they require a large number of adaptive queries to the target DL model.

## 1.1 Our Contribution

The general idea of iterative solvers is to approximate a (local) solution step-by-step when the global landscape of the objective function is unknown. The objective function in DL contexts is often highly complex, as it involves numerous factors, including the parameters of the neural network. Although there are attempts to analyze partial landscapes [11, 84, 57], understanding its entire landscape is nearly infeasible, and thus such iterative solvers may be the best approach until now. To overcome this fundamental limitation, we propose a novel method for non-adaptive adversarial face generation. Rather than embedding all aspects into the objective function and solving it iteratively, our approach interprets the feature space as much as possible and exploits its structural characteristics to refine the optimization problem. By leveraging this idea, we also show that our attack can be applied to a black-box setting where the adversary can obtain confidence scores from queries. Note that this scenario corresponds to attacking several real-world commercial face matching APIs, *e.g.*, provided by AWS [65] or Tencent [13]. With additional techniques tailored for this setting, we successfully generate adversarial faces for these APIs using scores from a single non-adaptive query composed of 100 faces. The adversarial faces generated from our attack on AWS CompareFace, along with the corresponding confidence scores from the API, are presented in Fig. 2. All these pairs surpass the API's default threshold of 0.8. More importantly, we emphasize that up to 13.7% of face pairs, consist of target face and adversarial face generated by our attack, surpasses the 0.99 confidence score—well

above the threshold suggested for law-enforcement according to Amazon's use-case guideline. These results demonstrate that our score-based non-adaptive approach can reliably generate adversarial faces even under realistic black-box constraints, highlighting the structural weaknesses of existing face recognition systems. For discussions on defensive strategies and responsible disclosure practices, please refer to Section 5.3 and the Ethics Statement 6.

## 2    Related Works

We briefly survey adversarial attack methods against FRSs. Similar to attacks on image classification [69, 53, 6], it is known that FRSs are vulnerable to adversarial attacks based on perturbations [19, 80, 48, 27]. In particular, recent studies have proposed attacks for black-box settings, successfully attacking real-world commercial FRSs, e.g., Tencent API [19] or Face++ [48, 27]. Along with these attacks, there is another branch of exploiting generative models for faces, e.g., StyleGAN [37], to craft adversarial examples [78, 82, 28, 68, 45, 63]. Most of these studies conducted transfer attacks via ensembles of the adversary's FRSs while employing the naturalness loss to ensure that the resulting adversarial faces are perceived as natural in humans' eyes. A series of works [82, 28, 68, 45] attempted to craft adversarial faces via makeups that were guided by the GAN-based image editing techniques or the aid of a vision-language model. Recently, [63] utilized a diffusion model and presented a method to weaken the diffusion purification effect. Nevertheless, we point out that all these attacks either require a huge number of adaptive queries [19] or heavily rely on the transferability. The former can be defended by detection methods for adaptive queries [9, 75], whereas the latter tends to exhibit a lower attack success rate.

## 3    Attributed Subsphere $\mathbb{S}_f^k$ and Non-Adaptive Adversarial Face Generation

### 3.1    Our Approach to Avoid Iterative Solvers using Attributed Subsphere Projection

The most DL-based FR model[2] are trained by so-called "*metric learning*" to make the feature space be like a metric space [51, 74, 16, 30, 55, 3, 39, 77, 33]. In particular, the above recent FR models utilize $(d-1)$-sphere $\mathbb{S}^{d-1}$ with angular distance metric $\mathtt{d}$ as a feature space. Assume that the target FR model is well trained by metric learning. Then, for any attribute $f$ (e.g., gender), we could naturally expect that the feature vector set $S_f$ of all images having $f$ lie close together in the feature space. Define the metric projection to $S_f$ as $p_{S_f}(\vec{x}) := \mathsf{argmin}_{\vec{y} \in S_f} \mathtt{d}(\vec{x}, \vec{y})$. For any $\vec{u} \in \mathbb{S}^{d-1}$, if we efficiently compute $p_{S_f}(\vec{u})$, then we may use it for adversarial face generation; for example, if $\vec{u}$ is a feature vector without $f$ and $\mathtt{d}(p_{S_f}(\vec{u}), \vec{u})$ is sufficiently small, then $p_{S_f}(\vec{u})$ is a feature vector of an adversarial example since it has attribute $f$ but $f$ is not present in the original image. Using well-known inversion methods that reconstruct faces from the corresponding templates extracted from the given FRS [54, 66, 67, 34, 59], we can recover the adversarial face image of $p_{S_f}(\vec{u})$. However, without assumptions on the structure of $S_f$, a naïve computation of $p_{S_f}$ becomes equivalent to exhaustive search, or it may require the use of generic iterative algorithms, e.g., gradient descent. To avoid such iterative methods, we establish a useful conjecture: the feature metric spaces $(\mathbb{S}^{d-1}, \mathtt{d})$ of all the metric-learning-based FR models share the following property.

**Conjecture 1.** *We call the attribute that most humans possess dominant attributes. (e.g., number of eyes, nose, and mouth.) There exist non-dominant attributes $f$ such that the feature vector set $S_f$ of all images having $f$ includes a $k$-sphere $\mathbb{S}_f^k$ with high probability $\mathrm{Pr}_{\vec{x} \in \mathbb{S}_f^k}[\vec{x} \in S_f]$. In addition, there exists an efficient algorithm to find a set of orthogonal unit vectors defining $\mathbb{S}_f^k$, called a basis.*

If the above conjecture is valid, we can use the projection to the $k$-sphere $p_{\mathbb{S}^k}$ instead of $p_{S_f}$ to efficiently compute without iterations. This is because a naïve projection to a $k$-sphere is rather straightforward by basic linear algebra; if we have a basis of $\mathbb{S}_f^k$, we can first project to $\mathbb{R}^k$ including the $\mathbb{S}_f^k$ and then normalize to be a unit vector. However, the remaining issue is whether $\mathtt{d}(p_{S_f}(\vec{u}), \vec{u})$ is sufficiently small. To address this, we present a proposition concerning the expected distance between a uniformly selected unit vector and its projection onto a subsphere.

**Proposition 1.** *Consider the metric space $(\mathbb{S}^{d-1}, \mathtt{d})$ and the metric projection to an arbitrary $k$-subsphere $\mathbb{S}^k \subset \mathbb{S}^{d-1}$, $p_{\mathbb{S}^k}(\vec{x}) := \mathsf{argmin}_{\vec{y} \in \mathbb{S}^k} \mathtt{d}(\vec{x}, \vec{y})$. Let $U$ be a uniformly chosen random variable*

---

[2]In this paper, FRS refers to the overall FR system whose final output is a score. We use the term "FR model" for "feature vector extractor", which does not contain the score computation process to avoid confusion.

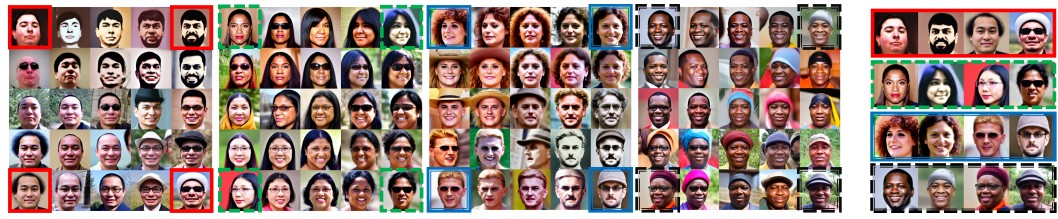

(a) Bilinear Interpolation on the Principal Components (PCs)          (b) PCs

Figure 3: The visualization of attributed subspheres $\mathbb{S}_f^k$ (left) from principal components (right).

*over $\mathbb{S}^{d-1}$ and $V := p_{\mathbb{S}^k}(U)$. Then, we have that the random variable $\cos^2(\mathtt{d}(U, V))$ follows the beta distribution* $\mathrm{Beta}(\frac{k}{2}, \frac{d-k}{2})$. *That is,* $\mathbb{E}[\cos^2(\mathtt{d}(U, V))] = \frac{k}{d}$.

Note that Prop. 1 provides the expectation of the squared cosine similarity, which may differ from the actual cosine similarity depending on the variances of $U$ and $V$. Although we experimentally verify that the expectation provides a sufficiently accurate approximation for our purposes, we defer both the proof and experimental validation to Appendix B due to space constraints. Importantly, the subsphere $\mathbb{S}^k$ is independent of the distribution of $U$. In our setting, if an attributed subsphere $\mathbb{S}_f^k$ is fixed and the adversary arbitrarily selects a target face image—regardless of $\mathbb{S}_f^k$—then there exists a feature vector in $\mathbb{S}_f^k$ whose average cosine similarity with the target is $\sqrt{k/d}$. To show its concrete implication, we note that the decision threshold of many FRS [16, 55, 3, 39] is typically set at most to $70°$, which corresponds to a cosine similarity of approximately $0.3420$. On the other hand, for $k = 128$ and $d = 512$, Prop. 1 gives $\sqrt{k/d} = 0.5$, i.e., $60°$ in angular terms. This means that for any 129-dimensional hyperplane, if we randomly select a face feature vector $u \in \mathbb{S}^{d-1}$ and compute its projection $\vec{v} \in \mathbb{S}^k$, then the reconstructed facial image corresponding to $\vec{v}$ would be recognized as the same identity as $\vec{u}$ by the aforementioned FRS, thus achieving our main objective.

## 3.2 Validation of the Existence of the Attributed Subsphere $\mathbb{S}_f^k$ and Conjecture 1.

Although the Proposition 1 and experimental results in Appendix B show the feasibility of our strategy to craft adversarial faces without iterative algorithms, it remains unclear whether Conj. 1 is indeed true, i.e., the existence of attribute-specific subspheres. Therefore, we now turn our attention to the $\mathbb{S}_f^k$ corresponding to attributes, e.g., race, or skin color, thus validating Conj. 1. To this end, we applied Principal Component Analysis (PCA), a classical algorithm for extracting representative bases (i.e., principal components) from a given distribution, to the set of feature vectors from faces sharing a specific attributes. Note that PCA is applied *in the deep feature space* (not in the pixel domain). We use PCA solely as an approximation to the basis of the attributed subsphere, not as a classical image-space preprocessor. We used the FairFace dataset [35], which provides nearly 110k annotated facial images, to collect samples labeled with selected attributes. Specifically, we selected four attributes: male, female, White, and Black. For each attribute, we ran PCA to obtain principal components and reconstructed facial images from the components using a pre-trained inverse model, Arc2Face [59]. These reconstructions are visualized and used in the subsequent analysis to evaluate whether the components span valid subspheres. In particular, by checking whether a linear combination (followed by normalization) of the principal components results in a facial image that still exhibits the same attribute as the original dataset used in PCA, we can empirically verify the existence of attributed subspheres. Fortunately, there is supporting evidence for this property: the semantic interpolation between facial images is known to be possible using inverse models by interpolating in the feature space [34, 66, 59]. Since a linear combination can be viewed as a sequence of linear interpolations, the subsphere spanned by principal components can be interpreted as a valid attributed subsphere. As shown in Fig. 3, although the pose may slightly vary, the interpolated facial images maintain identity coherence and preserve the intended shared attribute. To sum up, we conclude that Conj. 1 is indeed true, therefore the adversary can conduct an adversarial attack by exploiting attribute-specific spheres.

We remark that our argument is not about a specific choice of Arc2Face [59] we used, but about the inverse model itself of the metric learning-based FR model. Our argument still holds for other inverse models, such as NbNet [54]. Due to space constraints, we provide experimental results related to this aspect in Appendix D, as well as PCA results on other attributes or datasets and their interpolations.

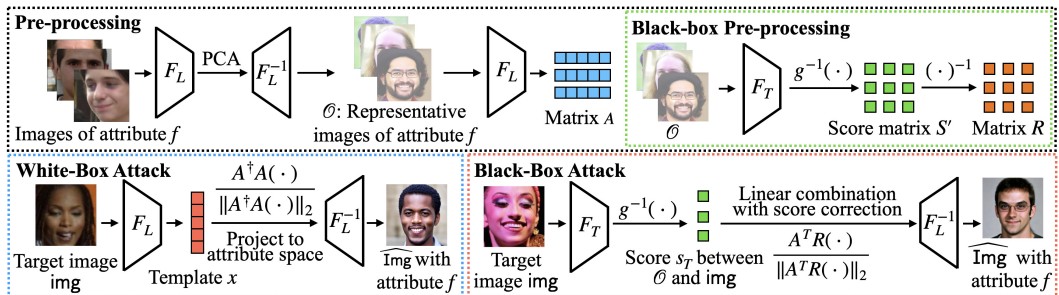

Figure 4: An overview of our adversarial face generation. In Figure, an attribute $f$ indicates male. $F_L : \mathcal{I} \to \mathbb{S}^{d-1}$ and $F_L^{-1}$ are adversary's own FR model and corresponding inverse model. $F_T : \mathcal{I} \times \mathcal{I} \to [0, 1]$ is target FRS whose output is confidence score and $g$ is a sigmoid function.

**Additional Note on Inverse Model Bias**. While our main results in Fig. 3 and Appendix D show that both Arc2Face [59] and NbNet [54] reconstruct faces consistent with the intended attributes, we observed that certain inverse models (e.g., [66]) exhibit systematic demographic bias, often reconstructing young white male faces regardless of the input feature vector. We emphasize that such bias originates from the inverse model architecture and training data rather than our projection mechanism. Consequently, although the projection step faithfully preserves the target attribute within the feature space, the perceptual quality of the reconstructed adversarial face may vary depending on the generative capability and bias of the chosen inverse model. We have explicitly noted this limitation in our final analysis and Appendix D.

### 3.3 Non-Adaptive Adversarial Face Generation

An intriguing property of adversarial examples is the transferability, where adversarial examples generated for one local FRS can deceive another target FRS. This property can convert "*white-box attacks*" to "*black-box attacks*". Therefore, we first present our adversarial face generation algorithm in a white-box setting, leveraging the insights discussed above. Let $\mathcal{I}$ be the ideal collection of all facial images, and let $\mathcal{I}_f \subset \mathcal{I}$ be the subset consisting of images with a specific attribute $f$. Given a facial image img $\in \mathcal{I} \setminus \mathcal{I}_f$, the goal of the adversary is to find another image $\overline{\text{img}} \in \mathcal{I}_f$ such that it is recognized as the same identity as img by a target FRS $T$. To achieve this, the adversary first runs PCA on $F(\mathcal{D}_f)$—where $\mathcal{D}_f$ is an $f$-attributed dataset and $F$ is the adversary's own FR model—to obtain a PCA matrix $M_f$ whose $i$-th row is $i$-th principal components denoted by $\vec{m}_{f,i}$. Using the inverse model $F^{-1}$, the adversary then reconstructs the corresponding facial images $O_i = F^{-1}(\vec{m}_{f,i})$. Next, the adversary defines a metric projection $p_{\mathbb{S}_f^k}(\vec{x}) : \mathbb{S}^{d-1} \to \mathbb{S}_f^k$ as $\frac{A^\dagger A\vec{x}}{\|A^\dagger A\vec{x}\|_2}$, where $A$ is the matrix whose $i$-th row is $F(O_i)$ and $A^\dagger$ is a pseudo-inverse of $A$. Then, the adversarial face $\overline{\text{img}}$ is generated as $F^{-1}(p_{\mathbb{S}_f^k}(F(\text{img})))$. Regardless of whether the adversary generates $\overline{\text{img}}$, such a sample always exists in the attributed subsphere and is close enough to img to be recognized as the same identity.

While the above (white-box) approach generates adversarial faces using $F$ and $F^{-1}$ alone, directly utilizing it for the transfer attack is insufficient for achieving a high attack success rate. Notably, we observe that some facial images consistently fail in transfer attacks, and this phenomenon of lower success rates is not limited to our attack but can also be observed with the classical adversarial attack method based on *iterative* solver. Due to space constraints, detailed analysis of such cases is provided in Appendix E. To overcome this, we extend our attack strategy by permitting the adversary to query the target FRS and exploit the obtained cosine similarity scores $\vec{s}$. To this end, we establish the following conjecture: there exists a *universal* basis $\mathcal{O}$ over facial images whose interpolations via a FR model and its inverse always produce similar images under the same coefficients, regardless of the choice of them. Our motivation is to view the metric projection function $x \mapsto \frac{A^\dagger A\vec{x}}{\|A^\dagger A\vec{x}\|_2}$ as a linear combination of rows of $A$; note that, by the definition of pseudo-inverse, $A^\dagger A\vec{x} = A^\mathsf{T}(AA^\mathsf{T})^{-1}A\vec{x}$, and $\vec{s} = A\vec{x}$. Hence, if we appropriately treat the $(AA^\mathsf{T})^{-1}$ term and the conjecture holds, then the adversary can produce the adversarial face by interpolating images in $\mathcal{O}$ through its FR model and its inverse with scores $\vec{s}$, which are obtained from querying img and images $O_i \in \mathcal{O}$.

**Conjecture 2.** *For $i \in \{1, 2\}$, let $F_i$ be well-trained FR model and $F_i^{-1}$ be its inverse. Then for any well-trained FRS $T$ with threshold $\tau_T$, there exists a set $\mathcal{O}$ of facial images s.t. for all $\vec{s} \in [-1, 1]^k$,*

$$T\left(F_1^{-1}\left(\frac{A_1^\mathsf{T}\vec{s}}{\|A_1^\mathsf{T}\vec{s}\|_2}\right), F_2^{-1}\left(\frac{A_2^\mathsf{T}\vec{s}}{\|A_2^\mathsf{T}\vec{s}\|_2}\right)\right) > \tau_T, \tag{1}$$

*where $A_i \in \mathbb{R}^{k \times d}$ is a feature vector matrix whose $j$-th row is $F_i(O_j)$ for $O_j \in \mathcal{O}$ and $j \in [k]$.*

Interestingly, we found that the $\mathcal{O}$ constructed from an $f$-attributed subsphere-as realized by PCA and the inverse model-does satisfy the required property in Conj. 2. To demonstrate this, for FR models $F_1$, $F_2$, and corresponding inverse models $F_1^{-1} = F_{1_A}^{-1}$, $F_2^{-1}$, respectively, we measured the distance of the feature vector of two images in Eq. (1) extracted from another FRS $T = F_3$. We sampled 10,000 score vectors $\vec{s}$ from the uniform distribution over $[-1, 1]^k$. For the choice of the image set $\mathcal{O}$, we considered three settings: (1) faces from our attribute-specific subsphere, (2) randomly sampled faces from the FairFace dataset, and (3) images consisting of uniformly sampled random pixels. The results are given in Fig. 5. We can observe that

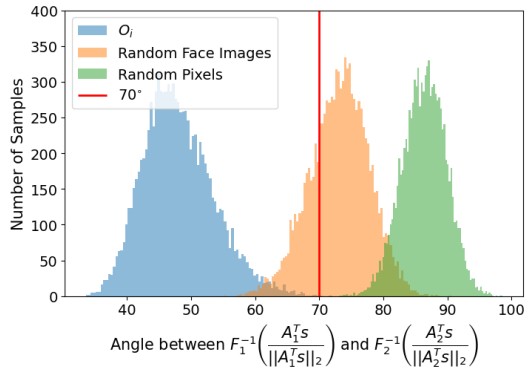

Figure 5: Angle histogram for Conj. 2.

the measured distances from the random pixel images are hardly within the threshold (red line), whereas a non-trivial number of those from faces lie within the threshold. This necessitates the condition in Conj. 2 that $\mathcal{O}$ should consist of faces. More importantly, we can figure out that almost all measured distances from the attribute-specific subsphere are within the threshold, whereas less than half of the randomly sampled faces lie outside the threshold. This indicates that our face image set $\mathcal{O}$ behaves well as the role of universal basis, therefore showing the validity of Conj. 2.

Building on Conj. 2, we can derive a formal relationship—particulary in the black-box setting—between the original image (without attribute $f$) and the crafted adversarial face image (with attribute $f$), by leveraging the projection $p_{\mathbb{S}_f^k}$ onto the $f$-attributed subsphere. In particular, to handle the $(AA^\mathsf{T})^{-1}$ term to connect the linear combination of rows of $A$ to the metric projection mapping, we introduce the *correction* matrix $R$. The equation is given as follows:

Subsphere projection: $p_{\mathbb{S}_f^k}(\vec{x}) = \frac{A^\dagger A \vec{x}}{\|A^\dagger A \vec{x}\|_2}$ and $\vec{x} = F_1(\text{img})$ $\qquad\qquad$ Conj. 2 and $R^{-1} = (A_1 A_1^\mathsf{T})^{-1}$

$$\text{img} \quad \approx F_1^{-1}\left(\frac{A_1^\dagger A_1 \vec{x}}{\|A_1^\dagger A_1 \vec{x}\|_2}\right) = F_1^{-1}\left(\frac{A_1^\mathsf{T}(A_1 A_1^\mathsf{T})^{-1}\vec{s}}{\|A_1^\mathsf{T}(A_1 A_1^\mathsf{T})^{-1}\vec{s}\|_2}\right) \approx F_2^{-1}\left(\frac{A_2^\mathsf{T} R^{-1}\vec{s}}{\|A_2^\mathsf{T} R^{-1}\vec{s}\|_2}\right),$$

face without attribute $f$ $\qquad\qquad A_1^\dagger = A_1^\mathsf{T}(A_1 A_1^\mathsf{T})^{-1}$ and $\vec{s} := A_1\vec{x}$ (Query) $\qquad\qquad$ adversarial face with $f$

From the above equations, $F_1$ is the target FR model that the adversary can query in a black-box, and $F_2$ is the FR model owned by the adversary. If we denote $\vec{s} := A_1\vec{x}$, then each component of $\vec{s}$ corresponds to the cosine similarity between the target facial image img and the images in $\mathcal{O}$. The left-hand side of the equation approximates a facial image that lies on the attributed subsphere and is close enough to img to be recognized as the same identity, regardless of whether it is explicitly generated by the adversary. Note that if the adversary had access to the full $F_1$ model, she could directly generate $\overline{\text{img}}$ using the white-box approach described earlier. However, since we can not access to the $F_1$, we proceed to reformulate the expression into a fully black-box compatible form. The second equality comes from the definition of the pseudo-inverse, namely, $A_1^\dagger = A_1^\mathsf{T}(A_1 A_1^\mathsf{T})^{-1}$. Here, if we denote $R = A_1 A_1^\mathsf{T}$ and $\tilde{s} := R^{-1}\vec{s}$, then we can observe that the numerator inside the $F_1^{-1}$ can be viewed as the linear combination of rows of $A_1$ with weights $\tilde{s}$. Hence, we can obtain the third equality by utilizing the Conj. 2. We can observe that all the involved values in the rightmost term in the equation are available to the adversary. $A_2$ can be locally calculated by the adversary and $\vec{s}$ can be obtained through queries. In addition, $R$ can also be obtained from $k^2$ cosine similarity scores by querying all the image pairs in $\mathcal{O}$. Note that $R$ is independent of the target image; the adversary can construct $R$ in advance before conducting the attack. Therefore, the adversary can craft the adversarial image by the formula in the rightmost term.

| Target Image | $F_{1_A}^{-1}$ : NbNet [54] | | | | | $F_{1_B}^{-1}$ : Arc2Face [59] | | | | |
| | [35]/Male | [35]/Female | [5]/White | [5]/Black | [5]/Asian | [35]/Male | [35]/Female | [5]/White | [5]/Black | [5]/Asian |
|---|---|---|---|---|---|---|---|---|---|---|
| Scores | 0.5782 | 0.6747 | 0.5278 | 0.6773 | 0.6295 | 0.5176 | 0.6367 | 0.4518 | 0.5533 | 0.6588 |
| Scores | 0.6154 | 0.4595 | 0.6791 | 0.5829 | 0.5248 | 0.5918 | 0.4522 | 0.6664 | 0.5020 | 0.4636 |

Table 1: Adversarial face examples using images from [29] (white-box setting; $\tau$ of $F_1$: 0.2432).

One caveat is that commercial FRSs typically do not return cosine similarity values, but rather confidence scores. Thus, we need an additional technique to convert the confidence scores into the cosine similarities. Fortunately, several methods have been proposed [43, 41], and we can directly adopt them for conducting our attack against commercial FRSs. Due to space constraints, we defer the detailed analysis of the correction matrix $R$ and the score transformation technique to Appendix F. Finally, we provide the full description of our black-box attack in Alg. 1.

---

**Algorithm 1** Projection (line 1-6) and Adversarial Face Generation (line 7-8)

---

**Require:** $f$-attributed dataset $\mathcal{D}_f$, a target face image img $\in \mathcal{I} \setminus \mathcal{I}_f$, a local FR model $F : \mathcal{I} \to \mathbb{S}^{d-1}$, its inverse model $F^{-1} : \mathbb{S}^{d-1} \to \mathcal{I}$, a target FRS $T : \mathcal{I} \times \mathcal{I} \to [0, 1]$, and hyperparameter $k \in [d]$

1: Run PCA on $F(\mathcal{D}_f)$ to obtain $M_f \in \mathbb{R}^{k \times d}$ whose row vectors are top-$k$ principal components
2: Set $O_i \leftarrow F^{-1}(\vec{m}_{f,i})$ for $\forall i \in [k]$, where $\vec{m}_{f,i}$ is $i$-th row vector of $M_f$
3: Set feature vector matrix $A \in \mathbb{R}^{k \times d}$, whose $i$-th row vector is $F(O_i)$ for $\forall i \in [k]$
4: Query and set $s'_{i,j} \leftarrow g^{-1}(T(O_i, O_j))$ for $\forall i, j \in [k]$, where $g(\cdot)$ is logistic sigmoid function
5: Set cosine similarity matrix $R \in [-1, 1]^{k \times k}$, whose $ij$-th component is $s'_{i,j}$ for $\forall i, j$
6: Define the projection to the $f$-attributed $k$-sphere by $p_{\mathbb{S}_f^k}(\vec{s}) := \frac{A^\top \vec{s}}{\|A^\top \vec{s}\|_2}$
7: Query and set $\vec{s} \in [-1, 1]^k$ whose $i$-th element is $g^{-1}(T(O_i, \text{img}))$ for $\forall i \in [k]$
8: **return** $\overline{\text{img}} \leftarrow F^{-1}(p_{\mathbb{S}_f^k}(R^{-1}\vec{s}))$

---

## 4 Experimental Results

### 4.1 Experimental Setting

We conducted evaluations on four face datasets: LFW [29], CFP-FP[64], and Age-DB[56], and the FairFace[35], which offers demographically balanced data to assess attack generalizability. We tested our attack using three open-source FRSs (resp. two commercial FRSs) with three inverse models. For obtaining an appropriate set $\mathcal{O}$ for Conj. 1, we extract attributed-specific PCA matrices ($k = 100$) using VGGFace2 [5] with annotations from [70] and annotated FairFace data. Thresholds $\tau$ were selected per dataset: accuracy-optimal values for Verification 3-sets and fixed thresholds for FairFace. Additional details and results for open-source FRSs and Tencent API are in Appendix A. For commercial FRSs, we used two thresholds provided by the corresponding service provider. Additional details for each model and results for open-source FRSs [16, 40] and Tencent API [13] is given in appendix, due to space constraints. For evaluating our adversarial face generation, we use the attack success rate (ASR). The ASR is the ratio of generated images that are both classified as the target identity and possess the target attribute and can be formulated as follows:

$$\text{ASR} = |\{\textstyle\sum_{i=1}^{|\mathcal{I} \setminus \mathcal{I}_f|} \mathbb{1}(T(\text{img}_i, \overline{\text{img}}_i) \geq \tau_\mathsf{T}) * \mathbb{1}(\overline{\text{img}}_i \in \mathcal{I}_f)\}| / |\mathcal{I} \setminus \mathcal{I}_f|,$$

where $\mathcal{I}$ represents the set of total images, $\text{img}_i \in \mathcal{I} \setminus \mathcal{I}_f$ refers to each individual target image, and $\mathbb{1}(\cdot)$ is a function mapping 1 if the input statement is true and 0 otherwise. To determine whether $\overline{\text{img}}_i \in \mathcal{I}_f$ in open-source and commercial target FRSs, we use an attribute classification model provided by FairFace [35] and corresponding APIs [65, 13], respectively.

## 4.2 Black-Box Attack

We first compare the ASR of the transfer attack[3] and the black-box attack with score queries in Tab. 2 using underline. To save space, we use M, F, W, B, and A to denote Male, Female, White, Black, and Asian, respectively. If the black-box attack performs better, it is underlined; otherwise, it is not. In most cases, the black-box attack with score queries outperforms the transfer attack without score queries in terms of ASR. We also present the effect of $R$ using colored text. The blue-colored text indicates the ASR with correction matrix $R$ is smaller than ASR without $R$. Since most of the ASRs are black-colored text, $R$ is effective. We now turn to a real-world black-box setting where an adversary can only obtain unknown metric scores. In Tab. 3, we present our ASR against the AWS CompareFace API [65] using gender-attributed $\mathcal{D}_f$. Our attack achieves significantly high ASR with a default threshold of 0.8. Even if we set the strict threshold of 0.99 recommended by Amazon for use cases involving law-enforcement, our attack achieves ASRs up to 13.70%. It is noteworthy that without matrix $R$, the ASR is only less than 1.5%. We also note that in the FairFace dataset, transfer attacks were not performed at all except for one case. Due to space constraints, we provide all ASR against Tencent CompareFace API [13] $F_T$ using race-attributed $\mathcal{D}_f$ in Appendix G.

| $F^{-1}$ | $f$ | Target Dataset | | | | | |
|---|---|---|---|---|---|---|---|
| | | LFW [29] | CFP [64] | AGE [56] | FairFace [35] | | |
| | | | | | $\tau_{\text{LFW}}$ | $\tau_{\text{CFP}}$ | $\tau_{\text{AGE}}$ |
| [54] $F_{1_A}^{-1}$ | M | 97.29 | 99.41 | 99.18 | 98.28 | 99.28 | 99.42 |
| | F | 93.33 | 96.02 | 97.44 | 94.84 | 96.63 | 96.75 |
| | W | 99.94 | 99.95 | 100 | 98.90 | 99.68 | 99.75 |
| | B | 87.98 | 90.61 | 88.66 | 92.12 | 92.63 | 92.68 |
| | A | 73.82 | 73.09 | 72.27 | 78.91 | 79.53 | 79.63 |
| [59] $F_{1_B}^{-1}$ | M | 70.82 | 86.66 | 91.16 | 81.31 | 91.94 | 93.63 |
| | F | 58.48 | 80.25 | 86.20 | 75.21 | 88.31 | 91.02 |
| | W | 84.42 | 90.43 | 94.84 | 73.32 | 87.96 | 90.49 |
| | B | 85.10 | 93.68 | 94.97 | 86.95 | 95.23 | 96.38 |
| | A | 73.97 | 84.90 | 89.81 | 83.69 | 90.95 | 91.98 |

Table 2: Black-box ASR on ViT-KPRPE [40] ($F_2$)

| $F^{-1}$ | Target | $f$ | with $R$ | | without $R$ | |
|---|---|---|---|---|---|---|
| | | | $\tau = 0.8$ | $\tau = 0.99$ | $\tau = 0.8$ | $\tau = 0.99$ |
| Transfer Attack without Queries | | | | | | |
| [54] $F_{1_A}^{-1}$ | [29] | M | N/A | | 47.96 | 7.06 |
| | | F | N/A | | 23.80 | 2.33 |
| | [35] | M | N/A | | 0.01 | 0 |
| | | F | N/A | | 0 | 0 |
| Direct Attack with Score Queries | | | | | | |
| [54] $F_{1_A}^{-1}$ | [29] | M | 91.45 | 5.95 | 71.75 | 1.49 |
| | | F | 86.46 | 4.51 | 60.19 | 0.82 |
| | [35] | M | 93.87 | 13.70 | 69.33 | 0.41 |
| | | F | 91.00 | 12.33 | 57.93 | 0.78 |
| [59] $F_{1_B}^{-1}$ | [29] | M | 79.55 | 0 | 37.92 | 0 |
| | | F | 66.62 | 1.23 | 24.35 | 0 |
| | [35] | M | 84.46 | 3.27 | 44.58 | 0.20 |
| | | F | 71.23 | 2.15 | 30.72 | 0 |

Table 3: Black-box ASR on AWS [65] ($F_A$)

# 5 Ablation Studies and Discussion

## 5.1 Comparison with Prior Work

Most prior works [78, 82, 28, 68, 45, 63] aim to protect a given face image—typically the adversary's own—by manipulating it so that the FRS classifies it as a different identity. In contrast, our work pursues the opposite objective: to generate synthetic facial images that are visually different from the adversary but are still recognized as the adversary by the FRS. This represents a fundamental difference: prior methods aim for *visual similarity with semantic difference*, while our method seeks *semantic similarity with visual difference*. Nevertheless, it is possible to adapt previous approaches to simulate our setting by reversing their direction: that is, by simply swapping the source image and the target image. It is worth noting that our method operates without source image. For comparison, we selected the most recent method [63], which follows a diffusion-based iterative attack paradigm, and re-purposed it in our setting. Since they use random face images from [61] as inputs without attribute constraints, we also generated adversarial faces targeting all attributes categories in our setting to ensure a fair comparison. In Tab. 4, we first provide the transfer ASR of both methods, the adversary has white-box surrogate models and attacks against AWS CompareFace. Then, we further evaluated our method by issuing queries to the actual API. Diffusion-based iterative methods typically operate in a white-box setting, requiring hundreds to thousands of adaptive queries and explicit gradient computations on the target FRS. In contrast, our non-adaptive approach relies solely on the reported similarity score and succeeds with a single query per target, demonstrating comparable effectiveness with far greater efficiency. We also present a visual comparison in Tab. 5, showing that our method generates facial images that are much more diverse and visually unrelated to the adversary, while still being classified as the same identity. On the other hand, the previous method tends to depend on the attribute of source image. Due to space constraints, additional details for Tab. 4 are provided in Appendix G.

---

[3]Transfer attacks are performed using white-box surrogate models and therefore incur no queries to the target FRS (query count = 0); the black-box results report attacks that query the target for similarity scores.

| Method | [63] | Ours (Transfer attack without queries) | | | | | Ours (Attack with 100 queries against AWS) | | | | |
|---|---|---|---|---|---|---|---|---|---|---|---|
| $f$ | [61] | Male | Female | White | Black | Asian | Male | Female | White | Black | Asian |
| $\tau = 0.8$ | 29.86 | 23.80 | 33.20 | 62.80 | 22.20 | 35.60 | 94.20 | 92.00 | 98.20 | 99.20 | 98.80 |
| $\tau = 0.99$ | 3.41 | 1.20 | 1.60 | 6.60 | 1.40 | 1.80 | 14.60 | 8.60 | 41.00 | 40.80 | 24.60 |

Table 4: ASR of [63] and ours evaluated on CelebA-HQ dataset [36] with different thresholds

| Target | [63] | | | | Ours | | | |
|---|---|---|---|---|---|---|---|---|
| | Source | Result | Source | Result | Transfer | Direct | Transfer | Direct |
|  |  |  |  |  |  |  |  |  |
| Scores | 0.0101 | 0.3557 | 0.0170 | 0.9964 | 0.6876 | 0.9792 | 0.9921 | 0.9936 |

Table 5: Visual comparison with [63]. The target image is shown on the left; ours used female and white attributed subspheres, respectively. To ensure a fair comparison, [63] used female and white source images. Additional results for male, black, and asian attributes are provided in Appendix G.

## 5.2 Black-box Attack on Non-facial Target

In Conj. 2, we did not impose any specific assumptions on the score vector $\vec{s}$, which indicates that the extraction of scores does not necessitate the input being facial images. In Tab. 23, we illustrated intriguing examples whose targets are non-facial images that provide some evidence that the proposed attack can be successfully performed not only on facial images but also on non-facial images, which are unrelated to the target model's task. For more details, please refer to Appendix G.

## 5.3 Possible Mitigation of Our Black-box Attack

We discuss possible mitigations against the proposed attack, focusing on the black-box setting, since in white-box or transfer scenarios, the adversary is assumed to have control over the FRS model. A straightforward defense would be to return only decisions (e.g., "accept"/"reject") instead of confidence scores. However, such an approach may violate regulations such as the EU AI Act [1] and GDPR [73], which mandate a right to explanation—usually realized via confidence scores. Therefore, we investigate defenses under the current threat model where confidence scores remain accessible. From a theoretical perspective, our attack is grounded in Prop. 1 and Conj. 2. The former enables exploiting feature subsphere to approximate target vectors; the latter enables improved ASRs using queried confidence scores. To mitigate Conj. 2, one option is to add noise to the returned score. While this may reduce FRS accuracy, it also lowers the ASR, thereby neutralizing the advantage over transfer-based attacks. To address Prop.1, we recall that the average cosine similarity between the original feature vector and its projection onto a $k$-dimensional subspace is $\sqrt{k/d}$. If $\tau$ is the threshold for a successful match, impersonation requires at least $k \geq d\tau^2$. Thus, increasing either $\tau$ or the dimension $d$ would raise the required number of queries. However, both trade-offs are not explored. Increasing $d$ imposes heavier computational and storage costs, especially for training. In fact, enlarging the dimension $d$ has not been actively studied and, as shown in the MFR benchmark[32], current models use only 128–1024 dimensions. Raising $\tau$, on the other hand, significantly lowers the TAR by increasing false rejections. Simply increasing $\tau$ on pre-trained models leads to severe performance degradation, making it unsuitable for practical deployment. To explore this direction more effectively, we implemented a prototype FRS trained from scratch with a higher $\tau$ as a proof-of-concept. As shown in Table 6, this configuration led to a significant reduction in ASR—by more than

| | $D_f$ | | | | |
|---|---|---|---|---|---|
| Target | Fair/Male | Fair/Female | VGG/White | VGG/Black | VGG/Asian |
| $F_2$ | 98.53 | 96.69 | 99.94 | 99.49 | 98.3 |
| $F_3$ | 99.62 | 98.55 | 100 | 99.68 | 99.29 |
| $F_P$ | 6.02 | 3.07 | 15.52 | 4.55 | 2.50 |

Table 6: To isolate the effect of varying acceptance thresholds independent of attributes, we report identity matching rates (IMR; see Supplementary Appendix G for definition). Reconstructions are obtained using $F_{1_A}^{-1}$ as the inverse model. $F_P$ denotes the prototype FRS, which shares the Inception ResNet-101 architecture with $F_3$ and is trained on the MS1MV3 [17] dataset.

80% compared to the ASRs against $F_2$ and $F_3$—while keeping the number of queries fixed. Further details are provided in Appendix G.

# 6 Ethics Statement

While our study introduces an effective attack method, its primary purpose is to provide a rigorous analysis that reveals structural weaknesses in FRSs and to promote the development of more robust and trustworthy recognition standards. Our work addresses a critical gap in FRS security by showing that non-matching attributes (e.g., gender or race) can still yield high cosine-similarity scores due to suboptimal threshold tuning (typically 0.2–0.3). This vulnerability can affect real-world identity-verification platforms, potentially enabling unauthorized access or impersonation. By exposing these flaws, our study informs service providers, regulators, and researchers of the need for stronger, attribute-aware defense mechanisms. For instance, our findings can help platforms adopt stricter verification thresholds or additional semantic consistency checks, thereby enhancing user safety and trust. Section 5.3 outlines practical defense strategies toward more secure systems.

To prevent misuse, we will not release the adversarial generation pipeline or related APIs. Benchmarking code for FRS evaluation will be shared under controlled access (e.g., to verified academic researchers via a private repository) to ensure responsible dissemination. Details specific to platform-level experiments are omitted in this paper to avoid potential misuse. These implementation details will be disclosed only to the affected service providers upon request, balancing transparency with risk mitigation. The core algorithm and non-sensitive experimental setup are fully described in the paper to ensure reproducibility for academic research.

Although our method can be conditioned on demographic attributes such as gender or race, this was not intended for discriminatory targeting. Instead, we demonstrate that even with distinct attribute values, adversarial faces can achieve high matching scores—highlighting structural vulnerabilities of existing FRSs rather than exploiting demographic bias. This motivates future work toward attribute-aware robustness and more secure, trustworthy face-recognition standards.

# 7 Conclusion

In this paper, we have investigated how close feature vectors with different attributes are in the feature metric space of FRS. To this end, we develop a process of exploring the feature space using *universal* basis based on Conj 2, which can serve as a compass to navigate in the dark, so that we could successfully extend this idea to non-adaptive adversarial attack in the black-box setting. This shows that although the metric learning, the dominant training method for FRS, provides a huge benefit for high accuracy, it is also useful for the adversary to design attacks with a high attack success rate. To the best of our knowledge, our attack is the first adversarial attack that is non-iterative, non-adaptive, and specialized to the metric learning-based DL technology. Rather than relying on gradients or perturbation constraints, our method leverages the intrinsic structure of the representation space to construct adversarial examples—challenging conventional assumptions about what is necessary for attack success. We leave some open questions, such as attacks specific to other DL-based systems trained in different ways than FRS. Another promising direction is to discover attribute-informed subspaces in a data-driven manner, for instance by using unsupervised or weakly supervised techniques such as clustering or latent-direction discovery. Such approaches could reveal more intrinsic structures of the feature space beyond manually defined attributes and further enhance both the theoretical and practical understanding of adversarial generation. In the opposite direction, it would also be interesting to investigate training paradigms that inherently reduce or restrict the exploitable structure for adversarial uses. Finally, we hope this study raises awareness of the vulnerability of commercial face APIs and encourages the development of secure and trustworthy face-recognition standards (e.g., ISO/IEC 24745).

# Acknowledgments

This work was supported by Culture, Sports and Tourism R&D Program through the Korea Creative Content Agency grant funded by the Ministry of Culture, Sports and Tourism in 2024(RS-2024-00332210)

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

# A  Additional Implementation Details

All experiments were conducted on a single NVIDIA A100 GPU using PyTorch [60]. In addition, when open-source face recognition models and their inverse models were utilized, the official inference codes provided by each model were used.

## A.1  Details of the Models Employed

In this section, we provide detailed descriptions of all the models used in our work. Specifically, we discuss the face recognition models, their inverse models, and the attribution classification models employed in our experiments.

**FRSs and their inverse models**  The specifications of the face recognition models for our attacks were briefly introduced in the main text. However, due to space limitations, we were unable to provide detailed information, including the model architectures, loss functions, and training datasets. Therefore, we present Tab. 7 which includes these details. For our experiments, $F_1$ was sourced from InsightFace [32], while the parameters for $F_2$ and $F_3$ were provided by CVLFace [4]. We trained $F_{1_A}^{-1}$ and $F_2^{-1}$ using the loss functions and training dataset detailed in Tab. 7. For $F_{1_B}^{-1}$, we utilized the parameters provided by Arc2Face [5]. Additionally, for the face recognition models, we show the thresholds at which the highest accuracies were achieved in evaluations on not only LFW but also CFP-FP and AgeDB, all within Tab. 8.

**Face Attribute Classification model**  We mentioned that to verify whether the results of our attack reflect the intended attributes, i.e., to check whether $\widehat{\mathsf{Img}_i} \in \mathcal{I}_f$, we utilized an attribute model. Specifically, we used a publicly available model from FairFace, which is known to distinguish gender and four racial groups (White, Black, Asian, and Indian). To evaluate the performance of this model, we compared the original labels from the FairFace Validation set with the outputs of the model in our experimental setup. The ACCs in Tab. 9 represent the percentage of images, for which the original

---
[4]https://github.com/mk-minchul/CVLface
[5]https://github.com/foivospar/Arc2Face

| Open-source FRS / Inverse Model | | | Train Dataset | TAR@FAR(%) | | |
|---|---|---|---|---|---|---|
| Notation | Architecture | Loss | Name | LFW | CFP-FP | AgeDB |
| $F_1$ | ResNet-100 | ArcFace | Glint360k [2] | 99.70@0.00 | 98.71@0.06 | 97.47@0.87 |
| $F_{1_A}^{-1}$ | NbNet-B | Perceptual | MS1MV3 [17] | N/A | N/A | N/A |
| $F_{1_B}^{-1}$ | Arc2Face | ID-conditioning | WebFace42m [88], FFHQ [37] | N/A | N/A | N/A |
| $F_2$ | Vit-KPRPE | AdaFace | WebFace12m [88] | 99.67@0.00 | 98.71@0.09 | 97.07@0.83 |
| $F_2^{-1}$ | NbNet-B | Perceptual | MS1MV3 [17] | N/A | N/A | N/A |
| $F_3$ | Inception ResNet-101 | ArcFace | WebFace4m [88] | 99.73@0.07 | 98.74@0.20 | 97.33@1.17 |
| $F_A$ | AWS CompareFaces API | | | | | |
| $F_T$ | Tencent CompareFace API | | | | | |

Table 7: Description of Open-Source Face Recognition Systems (FRSs) and Their Inverse Models.

**Note on Inverse Model Families**. Among the inverse models used in our experiments, Arc2Face [59] is a diffusion-based model that reconstructs high-fidelity faces from feature vectors, while NbNet [54] represents a GAN-based deconvolutional inverse network. Although our main results focus on Arc2Face and NbNet for consistency, the proposed framework is fully compatible with other inverse architectures such as GAN-based models (e.g., Vec2Face [20]). This underscores the generality of our approach across different inverse model families, as long as the model can reliably map feature vectors back to the pixel domain.

| Dataset | LFW | CFP-FP | AgeDB |
|---|---|---|---|
| $F_1$ | 0.2432 | 0.2092 | 0.1832 |
| $F_2$ | 0.2272 | 0.1892 | 0.1772 |
| $F_3$ | 0.2212 | 0.1832 | 0.1652 |
| $F_A$ | 0.8 (Default Threshold) | | |
| $F_T$ | 0.6 (Default Threshold) | | |

Table 8: Thresholds for FRSs Across Face Verification Datasets (LFW, CFP-FP, AgeDB).

label matches the attribute, that were correctly classified by the model. More specifically, in the case of FairFace, East Asians and South Asians were both considered as Asians.

| $f$ | Male | Female | White | Black | Asian |
|---|---|---|---|---|---|
| $|I_f|$ | 5792 | 5162 | 2085 | 1556 | 2965 |
| ACC | 95.7 | 96.07 | 93.72 | 94.6 | 96.93 |

Table 9: Performance of attribute classification models: The first row represents the attributes $f$, the second row shows the number of images in the FairFace validation set that the original label equals with $f$, and the last row shows the accuracy.

## A.2 Statistics of the Datasets

This subsection presents the statistics of the image datasets targeted in the experimental attacks conducted in this study.

**Target Image Datasets.** We utilized both facial images and non-facial images as targets, with their respective statistics summarized in Tab. 10.

| | Facial | | | | Non-Facial | | |
|---|---|---|---|---|---|---|---|
| Dataset | LFW | CFP-FP | AgeDB | FairFace | CIFAR-10 | Flower-102 | Random |
| Imgs/(IDs) | 13,233/5749 | 7,000/500 | 16,488/568 | 10,954/ N/A | 10,000 | 6,149 | 10,000 |

Table 10: Statistics of the attack target datasets (both of facial and non-facial).

Also, when we perform our proposed attack, target images have to be chosen as images that don't possess the target attribute. So we provide Tab. 11 which presents the number of images that possess attributes $f$. The attribute judgment for constructing this table was also carried out using the attribute classification model mentioned earlier. Of course, when original labels are available, such as in the case of FairFace, the original labels were used for classification. Specifically, original labels with East Asians and South Asians were both considered as Asians in the FairFace case.

**Dataset for $D_f$.** To perform our attack, it is necessary to extract the PCA matrix from the dataset $D$ with $f$-attribute. Therefore, we created $D_f$ using the attribution labels provided by FairFace and MAADFace about the VGGFace2 dataset. We selected five attributions: two for gender from FairFace and three for race from VGGFace2. The number of images per attribute is provided in Tab. 12.

| Dataset | $f$ | | | | |
| --- | --- | --- | --- | --- | --- |
| | Male | Female | White | Black | Asian |
| LFW | 9,341 | 2,659 | 4,286 | 3,322 | 1,755 |
| CFP-FP | 10,433 | 3,567 | 7,604 | 3,442 | 1,546 |
| AgeDB | 7,259 | 4,741 | 5,568 | 3,559 | 733 |
| FairFace | 5,792 | 5,162 | 2,085 | 1,556 | 2,965 |
| CIFAR-10 | - | - | - | 1,264 | - |
| Flower-102 | - | - | - | 445 | - |
| Random | - | - | - | 3,324 | - |

Table 11: Statistics of target datasets for $I_f$.

| $D$ | $f$ | Imgs |
| --- | --- | --- |
| FairFace | Male | 45,986 |
| | Female | 40,758 |
| VGGFace2 | White | 2,136,057 |
| | Black | 157,109 |
| | Asian | 115,021 |

Table 12: Statistics of datasets for $D_f$.

# B  Validation of Proposition 3.1.

In this section, we provide an omitted proof for Proposition 1. We also experimentally verify Proposition 1 by measuring distances between feature vectors and random $k$-hyperplane or k-hyperplanes derived from faces whose feature vectors are almost orthogonal to each other.

*Proof of Proposition 3.1..* Let us denote $\mathcal{P}_k$ as the $k$-hyperplane containing $\mathbb{S}^k$ and define a random variable $\widetilde{V}$ as a projection of $U$ onto $\mathcal{P}_k$. Then we have that $\cos^2\left(\mathsf{d}(U,V)\right) = \|\widetilde{V}\|_2^2$. Hence, we focus on analyzing $\|\widetilde{V}\|_2^2$ instead of $\cos^2\left(\mathsf{d}(U,V)\right)$.

Because of the radial symmetry of the hypersphere, the distribution of $\widetilde{V}$ is identical to the following random variable $W = (W_1, \ldots, W_d)$ defined over $\mathbb{R}^d$:

$$W_i = \begin{cases} U_i & \text{if } i \leq k \\ 0 & \text{otherwise.} \end{cases},$$

where $U_i$ is the $i$'th component of $U$ for $i \in [d]$.

To analyze $W$, we first note that for a random variable $X = (X_1, \ldots, X_d) \sim N(0, I_d)$, the random variable $Z := \frac{X}{\|X\|_2}$ follows the uniform distribution over $\mathbb{S}^{d-1}$. That is, analyzing the distribution of $\|\widetilde{V}\|_2^2$ is equivalent to

$$\|\widetilde{V}\|_2^2 \stackrel{d}{\cong} \|W\|_2^2 \stackrel{d}{\cong} \frac{\sum_{i=1}^k X_i^2}{\sum_{i=1}^k X_i^2 + \sum_{j=k+1}^d X_j^2} \tag{2}$$

where $\stackrel{d}{\cong}$ means that two random variables are equivalent in terms of distribution.

Here, we can observe that each $X_i, X_j$ for $i \neq j$ are pairwise disjoint. In addition, $\sum_{i=1}^k X_i^2$ and $\sum_{j=k+1}^d X_j^2$ follow chi-squared distribution with degree of freedom $k$ and $d-k$, respectively. That is, the RHS of Eq. (2) follows $\text{Beta}(\frac{k}{2}, \frac{d-k}{2})$ by definition. By using the fact that the mean of the $\text{Beta}(\alpha, \beta)$ is $\frac{\alpha}{\alpha+\beta}$, we finally obtain $\mathbb{E}[\|\widetilde{V}\|_2^2] = \frac{k}{d}$. This completes the proof. $\qquad \square$

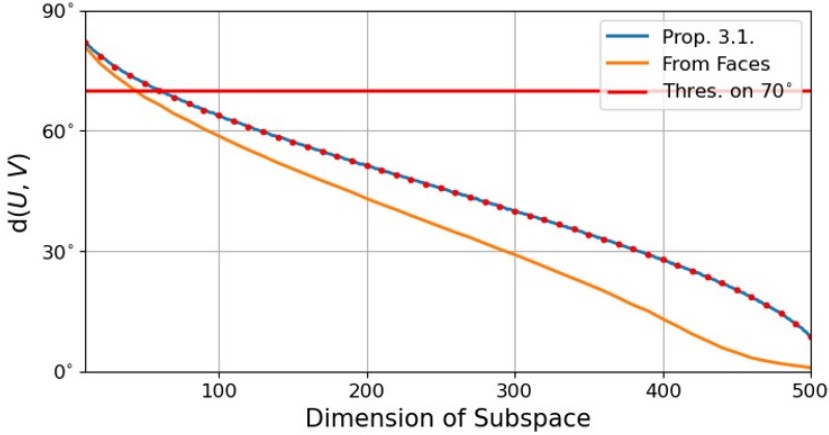

Figure 6: Measured distance $\mathsf{d}(U, V)$ of the projected face feature vector onto the subsphere. Red dots indicate the theoretically predicted value according to Prop. 1.

With an inverse model, we can find a set of facial images corresponding to the basis of the given subsphere. Using them, we attempted to simulate the settings and result of Prop. 1. First, for a randomly selected $k$-subsphere $\mathbb{S}^k$, we measured $\mathsf{d}(x, p_{\mathbb{S}^k}(x))$ for the given feature vector $x \in \mathbb{S}^{d-1}$. All feature vectors are extracted from the merge of the LFW, CFP-FP, and AgeDB datasets, excluding overlapping images. In addition, we also measure the same quantity under the same setting as above, except we to sample a $k$-subsphere from faces whose feature vectors are *almost* orthogonal to each other. We can view this as sampling a feature subsphere with considering the distribution of facial images, rather than independently from it. Such a set of faces is called an orthogonal face set (OFS), which was first proposed by [41]. To generate them, we devised and exploited an efficient algorithm by utilizing the input space of the inverse model, whose description is given in Appendix A. We used the pre-trained ArcFace [16] as the FR model to obtain feature vectors. For the inverse model, we used the pre-trained NbNet [54] of the aforementioned FR model. The detailed description of each model is given in Tab. 7 as $F_1$ and $F_{1_A}^{-1}$, respectively.

The results are illustrated in Fig. 6. From this figure, we can observe that the simulated result (blue line) well coincides with the theoretically predicted value via approximation (red dots). More importantly, we can observe that both the distance calculated from the OFS (orange line) and the blue line lie below the red line corresponding to $70°$ when $k \geq 100$, *i.e.*, such a choice is sufficient for generating a face that is identified as the same person with the target identity. One can figure out that the simulation result from faces is strictly less than that from Prop. 1 for all dimensions of the subsphere. This result indicates that there exist *good* feature subspheres to obtain a more accurate feature vector than a uniformly selected one, and more importantly, one way to obtain them is to select a feature subsphere derived from actual facial images.

## C  Efficient OFS Generator

For generating OFSs, Kim *et al.* [41] utilized a rather naïve approach by collecting lots of facial images and finding a subset being an OFS. However, their method is not scalable because the possible number of subsets is exponentially many with respect to the size of the desirable OFS set, and more importantly, there is no guarantee whether such an OFS exists in a pre-selected set of facial images.

To mitigate these issues, we propose an alternative approach by optimizing on the latent space of the inverse model. More precisely, instead of searching an OFS over the facial images, we focus on finding the set of latent vectors $\{z_1, \ldots, z_k\} \subset \mathbb{S}^{d-1}$ of the inverse model $F^{-1} : \mathbb{S}^{d-1} \to \mathcal{I}$ of a FRS $F : \mathcal{I} \to \mathbb{S}^{d-1}$. Note that the inverse model does not give an *exact* inverse, so we need to ensure that the feature vectors corresponding to facial images $\{F^{-1}(z_1), \ldots, F^{-1}(z_k)\}$ are orthogonal to each other. Thus, if we denote $Z \in \mathbb{R}^{k \times d}$ as a matrix whose row vectors consist of $\{z_1, \ldots, z_k\}$ and $\mathcal{W} = F \circ F^{-1} : \mathbb{S}^{d-1} \to \mathbb{S}^{d-1}$ as a sequential composition of $F^{-1}$ and $F$, then we can formulate the optimization problem for finding an OFS as follows.

$$Z^* = \arg\min_{Z} \|\mathcal{W}(Z)\{\mathcal{W}(Z)\}^T - I_k\|_F$$

---

**Algorithm 2** Efficient OFS Generator

---

**Require:** A FRS model $F : \mathcal{I} \to \mathbb{S}^{d-1}$ and its inverse $F^{-1} : \mathbb{S}^{d-1} \to \mathcal{I}$, the size of OFS $k \in [d]$, and a learning rate $\alpha \in \mathbb{R}_{>0}$.
**Ensure:** A set of facial images $\mathcal{O} \subset \mathcal{I}$ being an OFS and $|\mathcal{O}| = k$.
 1: Initialize $\{z_1, \ldots, z_k\} \leftarrow \mathcal{U}(\mathbb{S}^{d-1})$ and set a matrix $Z \in \mathbb{R}^{k \times d}$ whose $i$'th row is $z_i$ for $i \in [k]$.
 2: **while** Not Converged **do**
 3:     Compute $\mathcal{O} \leftarrow F^{-1}(Z)$ and $\widehat{Z} \leftarrow F(\mathcal{O})$.
 4:     Compute $L \leftarrow \|\widehat{Z}\widehat{Z}^T - I_k\|_F$
 5:     Update $Z \leftarrow \mathsf{Normalize}(Z - \alpha \cdot \frac{\partial L}{\partial Z})$
 6: **return** $F^{-1}(Z)$.

---

where $I_k$ denotes the $k \times k$ identity matrix and $\| \cdot \|_F$ denotes the Frobenius norm for matrices. We can solve this problem via projected gradient descent with a constraint that each row vector belongs to $\mathbb{S}^{d-1}$. We select the initial $\{z_1, \ldots, z_k\}$ as a random sample from the uniform distribution $\mathcal{U}(\mathbb{S}^{d-1})$ over the $\mathbb{S}^{d-1}$, which can be implemented by normalizing vectors sampled from the Gaussian distribution $N(0, I_k)$. For simplicity, we denote Normalize as an operator that normalizes the row vectors of the given matrix to be unit vectors. We summarize the above idea as Algorithm 2.

In our experiment for producing Fig. 6, we used the Adam optimizer [42] to solve the optimization problem, selecting $\alpha = 0.1$. In addition, we terminate the algorithm when it has not converged after 100 updates. We also remark that for a large $k$, *e.g.*, $k > 256$, the algorithm may not converge within 100 iterations. We suspect that this is because the face feature vectors would not occupy the whole hypersphere; only a subsphere would correspond to actual faces. There has been some evidence for this phenomenon, such as studies on the dimensionality reduction techniques for face templates [22, 21]. Nevertheless, our algorithm is sufficient for our purpose, and we leave more analysis on this aspect as future work.

## D  Additional Examples of Attribute-Specific Subspheres

We provide more examples of attribute-specific feature subspaces for analyzing the validity of Conj. 1.

### D.1  More Finer Interpolation Results

Although the example in Fig. 3a is sufficient for our purpose, because of the space limit, the interpolations were done rather coarsely. To complement this, we also provide the $10 \times 10$-sized attribute-specific subspheres generated from the same algorithm as Sec. 3.3. The visualization result is given in Fig. 7. Similar to Fig. 3a, we can observe that each image contained in the subsphere still shares the common attribute, while the deviation between adjacent images is reduced because of the finer interpolation.

### D.2  Interpolation from Other Attributes

We note that the FairFace dataset or VGGFace datasets provide more attributes than we experimented with, *e.g.*, more races such as Asian or Middle Eastern, attributes about age, or accessories such as glasses or baldness. In this section, we provide more interpolation results about them to investigate the *non-dominant* features satisfying the Conj. 1.

We selected the following attributes from each dataset: In FairFace, we selected Asian, Indian, and Latino-Hispanic for races, and ages range from 0-9, 20-39, and 50 or older. On the other hand, in VGGFace, we selected accessories having a hat, glasses, or baldness. We note that the size of all collected images across attributes is more than 9,000. We used the same FR model and its inverse model as the previous experiment.

We provide the facial images corresponding to each subsphere in Fig. 8, Fig. 9, and Fig. 10, respectively. From these figures, we can observe that each subsphere largely catches the desired attributes, and interpolations between adjacent images seem to be done smoothly. However, for subspheres regarding ages in Fig. 9, we can observe that some images in the range 0-9 would not fit with their

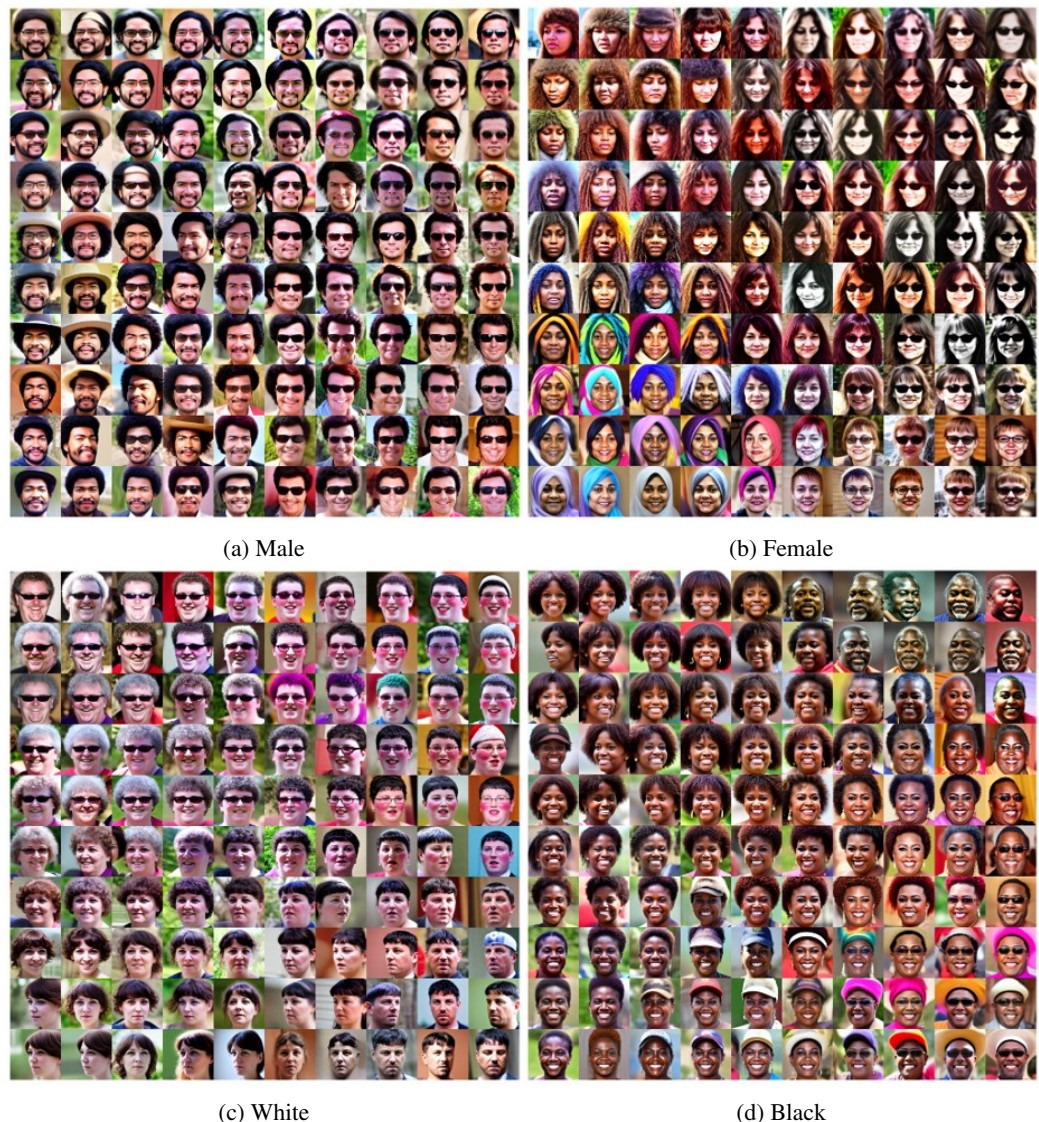

(a) Male

(b) Female

(c) White

(d) Black

Figure 7: Examples of attribute-specific subspheres with a finer interpolation ($10 \times 10$-sized).

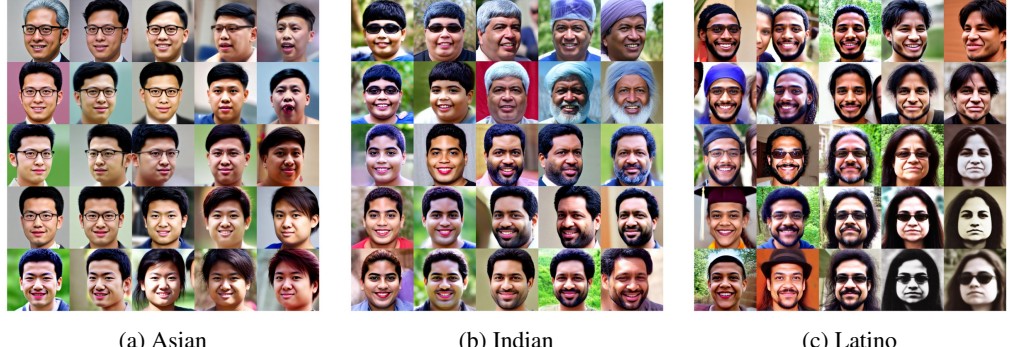

(a) Asian

(b) Indian

(c) Latino

Figure 8: Attribute-specific subspheres from more types of races.

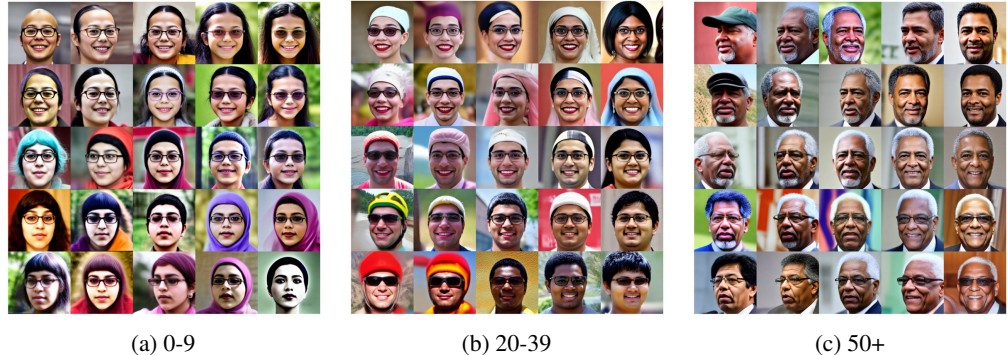

| (a) 0-9 | (b) 20-39 | (c) 50+ |

Figure 9: Attribute-specific subspheres from various range of ages.

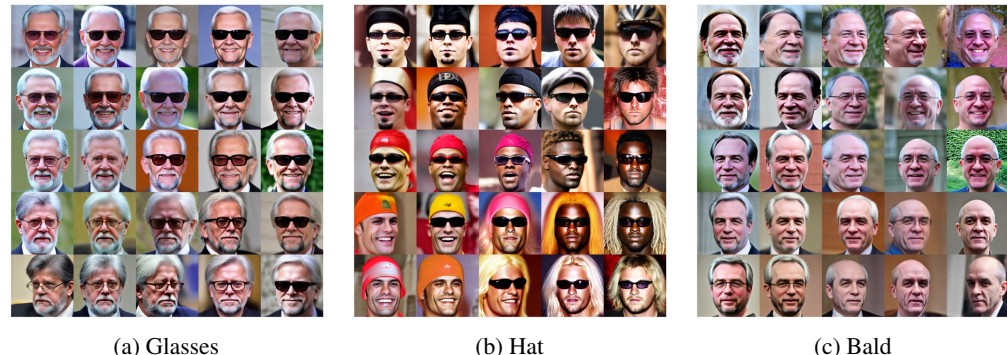

| (a) Glasses | (b) Hat | (c) Bald |

Figure 10: Attribute-specific subspheres from various accessories.

attribute, though we can observe that the faces seem to get older as being placed in right. We guess the reason for this phenomenon is two-fold: first, the FR model and its inverse would not learn much about faces in the 0-9 age range. In fact, as provided in the original paper of WebFace42M [88], which is the training dataset of the Arc2Face model we used, the age distribution of the training dataset is concentrated on ages more than 20. Hence, we suspect that the FR model and its inverse model would not be familiar with handling faces within this 0-9 regime.

On the other hand, we also provide another interpretation by considering how much the attribute age contributes to extract a discriminative feature in terms of identities. As we can see in the benchmark results of datasets testing the ability of the FR model to handle the variation in age, including AgeDB [56] or CALFW [87], recent FR models, including the model in our experiment, achieve a good accuracy on these benchmark datasets. That is, we can expect that varying the age would not lead to a huge derivation on the feature vector, and thus failing to form a subsphere because of the collapsing effect of the FR model on faces with the same identity but different ages. From this argument, we further infer that such a phenomenon would occur for other attributes that would not play an important role in extracting identity-specific features. As evidence for this, note that a similar phenomenon occurs at the subsphere corresponding to hats, while this does not occur for other accessories, *e.g.*, glasses or bald. We think that these factors complexly affected the production of these non-trivial results, and we leave further analyses about them and the effect of these attributes on our attack as interesting future work.

### D.3 Subspaces from Other Inverse Models

To show that our results in Sec. 3.3 are regardless of the choice of the inverse models, we also provide the results of subspaces from other inverse models, including NbNet [54] and the StyleGAN-based inverse model proposed by Shahreza and Marcel [66]. For NbNet, we used the same model as $F_{1_A}^{-1}$ in our experiments. On the other hand, for the inverse model by [66], we utilized their official implementation with a public pre-trained model. The details about the latter model will be found in their original paper. We conducted the same experiments as in Sec. 3.3.

The results are given in Fig. 11 and Fig. 12 for each model, respectively. From this figure, we can observe that the subspheres made from NbNet showed the desired result, while almost all faces from [66] were alike to white males. We guess that this is because their inverse model uses unnormalized features as an input, so their inverse model is not compatible with metric learning-based FR models using cosine similarity. In fact, we observed that the output image varies as we change the norm of the feature vector, so we multiplied the mean norm of the feature vector for each principal component. In addition, the authors of this inverse model reported that their inverse model struggled to invert facial images from some attributes, *e.g.*, Asian, Black, or oldness. This result also indicates that the capability of the inverse model with respect to the diversity of the generated faces is also crucial for conducting our attack, especially for realizing the selected attribute of the adversarial face.

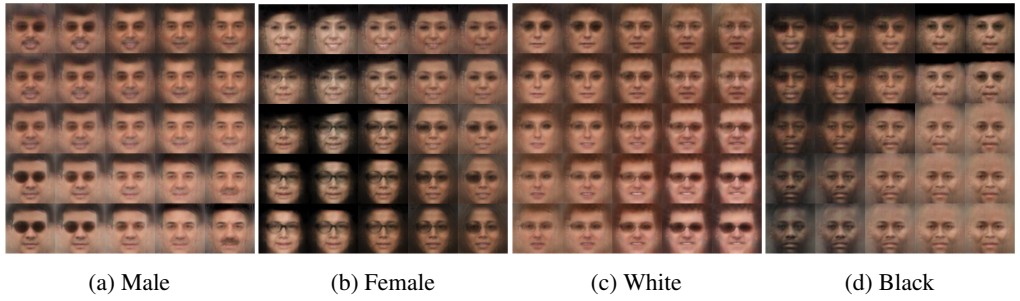

| (a) Male | (b) Female | (c) White | (d) Black |

Figure 11: Attribute-specific subspaces from NbNet.

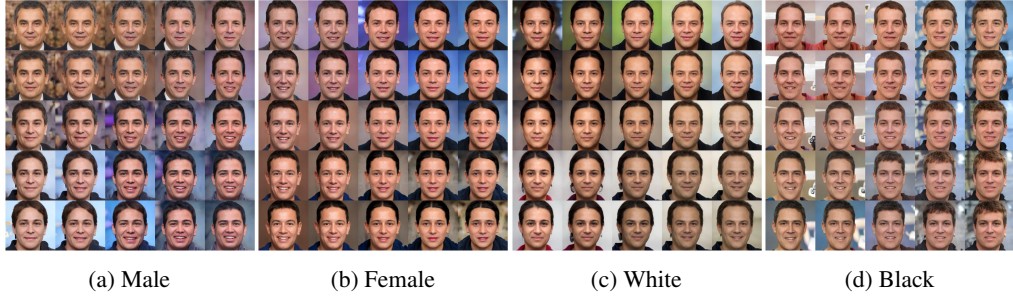

| (a) Male | (b) Female | (c) White | (d) Black |

Figure 12: Attribute-specific subspaces from [66].

# E    Transfer Attack (Naïve Approach)

An intriguing property of adversarial examples is the transferability, where adversarial examples generated for one local FRS can deceive another target FRS, often with a different architecture. This intriguing property can convert *"white-box attacks"* to *"black-box attacks"* that can circumvent the need for detailed knowledge of the target FRS. Similarly, we can expect that the adversarial faces generated by our attack in the white-box setting may deceive another target FRS. The corresponding experimental result on the LFW dataset is illustrated in Fig. 13. Although these transfer attacks show some success rates, they have fundamental limitations unless they do not use information from the target FRS. For example, the FR model trained from a strongly biased dataset suffers from inconsistent accuracy, and we cannot expect the transfer attack to work well if the target image of the adversarial example we generated is from a long-tailed distribution, as the similarity in the image pair is low. To support this argument, we illustrated a histogram of angular distances between original target images and corresponding adversarial images from $F_{1_A}^{-1}$ in another target FRS $F_2$ using the FairFace [35] dataset in Fig. 13, which is significantly lower than that using the LFW dataset. Note that while more than 75% of MS1MV3, which is a representative public training dataset, or LFW datasets, are comprised of Caucasian face images, FairFace is a dataset comprised of more than 75% non-Caucasian face images.

Notably, this phenomenon of lower transfer attack rates is not limited to our attack but can also be observed in adversarial examples generated by the classical white-box adversarial attack method

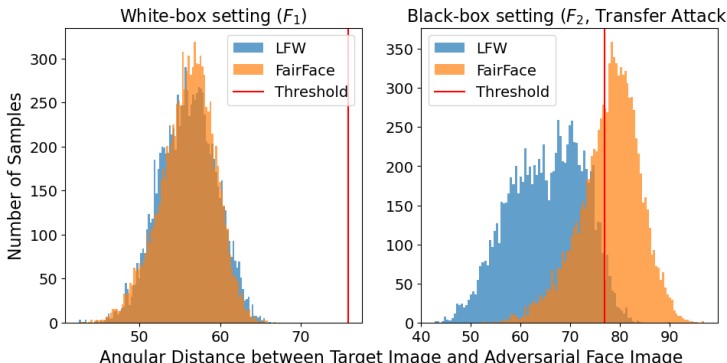

Figure 13: Angular distance histograms. An adversarial face is accepted if its angular distance is below the threshold.

| Noise Bound | $\epsilon = 0.25$ | | $\epsilon = 0.75$ | |
|---|---|---|---|---|
| Dataset | LFW | FairFace | LFW | FairFace |
| White-box | 26.4% | 89.9% | 99.8% | 99.8% |
| Transfer | 6.3% | 1.0% (-88.9%) | 87.6% | 5.3% (-94.5%) |

Table 13: Comparison of attack success rates in different settings. Note that we only performed the untargeted attack to ensure a fair comparison, since the attack success rate is affected by the starting image in targeted attack. Note that we use $\epsilon$ as the $\ell_2$ norm bound for PGD attack with 20 iterations.

based on *iterative* solver. In Tab. 13, we record the white-box and black-box (transfer) attack success rates from the PGD algorithm using the same two datasets. As with our attack, there are still large gaps regardless of noise bound. That is, it occurs because two FR models do not share similar metric spaces around these difficult samples, and we intend to propose an attack that covers even these difficult samples. Note that there are other ways to improve transferability, such as using surrogate models that mimic the behavior of the target system. However, we do not consider such methods because training a surrogate model requires a huge number of queries and is impractical when the target FRS is a real-world application face API.

## F Additional Techniques

In this section, we introduce some techniques to proceed with our attack.

**Confidence to Cosine Similarity.** Our first technique is a method to transform confidence scores into cosine similarity scores, which can then be applied to our attack when the target FRS is API. A recent study by [43] proposed a method for calculating confidence scores in FRS by assessing the cosine similarity between two input images. Their method deploys the DOGBOX algorithm [72] to determine the coefficients of a logistic sigmoid function $g(s) = \frac{L}{1+e^{-k \cdot (s-d_0)}} + b$, where $L, d_0, k$, and $b$ are coefficients that need to be fitted. In [41], which proposed a reconstruction attack on commercial FRS, they followed the method of [43]. We note that they used only true-false image pairs that are different from the images used in actual queries when generating adversarial face images. Instead, we determined the coefficients using the images used in queries. Precisely, we fitted each coefficient by the following objective function:

$$\underset{L,d_0,k}{\arg\min} \sum_{O_i \in \mathcal{O}} \sum_{O_j \in \mathcal{O}} |g(\langle F(O_i), F(O_j) \rangle) - T(O_i, O_j)|,$$

where $F : \mathcal{I} \to \mathbb{S}^{d-1}$ is the local FR model and $\mathcal{O}$ is a set of facial images defined in Conj. 2.

**Additional Technique: Correction Matrix.** We now present our second technique, called correction matrix, which converts scores to an appropriate form based on Conj. 2. We provide our analysis about why such a transformation is indeed effective. By the definition of a pseudo-inverse matrix, we easily obtain the equality $A_1^\dagger A_1 x = A_1^\mathsf{T} (A_1 A_1^\mathsf{T})^{-1} A_1 x$. If our inverse model $F^{-1}$ is the exact

inverse function of $F$, we obtain $(A_1 A_1^\top) = I_k$ where $I_k$ is the $k$-dimensional identity matrix. This is because $M_{f,i}$ are top-$k$ principal components of the PCA matrix and then orthogonal to each other for all $\forall i \in [k]$. However, since our inverse model $F^{-1}$ is an approximated version, we multiply the correction matrix $R = (A_1 A_1^\top)^{-1}$ by $\vec{s}$ to obtain a more accurate approximation for Conj. 2. Note that Conj. 2 applies to the same $\vec{s}$ and transpose matrix $A_i^\top$ of $A$, not its pseudo-inverse $A_i^\dagger$. Thus, here the correction matrix $R$ transforms to the same $\vec{s}$ in terms of Conj. 2, providing a bridge between the $F_1^{-1}\left(\frac{A_1^\top \vec{s}}{\|A_1^\top \vec{s}\|_2}\right) \approx_T F_2^{-1}\left(\frac{A_2^\top \vec{s}}{\|A_2^\top \vec{s}\|_2}\right)$ and the final generating term of the white-box attack defined using the pseudo-inverse, $\widehat{\text{img}} = F_1^{-1}\left(\frac{A_1^\dagger A_1 x}{\|A_1^\dagger A_1 x\|_2}\right)$.

# G    Addditional Experiments

## G.1    White-box Attack Setting

To evaluate the effectiveness of the white-box setting, we first report the attack success rates against the FR model $F_1$. These results are summarized in Tab. 14. As expected, since the adversary has full access to the model architecture and parameters, the attack achieves consistently high success rates, mostly above 90%, with a minimum above 80%, across all attributes and inverse models.

| $F^{-1}$ | $\mathcal{D}_f$ | Target Dataset | | | | | |
|---|---|---|---|---|---|---|---|
| | | LFW | CFP | AGE | FairFace (FF) | | |
| | | | | | $\tau_{\text{LFW}}$ | $\tau_{\text{CFP}}$ | $\tau_{\text{AGE}}$ |
| $F_{1_A}^{-1}$ | FF/Male | 95.83 | 94.31 | 96.79 | 93.14 | 93.37 | 93.47 |
| | FF/Female | 91.29 | 89.02 | 92.41 | 89.92 | 90.14 | 90.14 |
| | VGG/White | 99.74 | 99.34 | 99.94 | 94.07 | 94.08 | 94.08 |
| | VGG/Black | 96.35 | 95.15 | 97.03 | 90.76 | 90.81 | 90.83 |
| | VGG/Asian | 83.45 | 83.41 | 80.24 | 90.22 | 90.3 | 90.3 |
| $F_{1_B}^{-1}$ | FF/Male | 92.25 | 93.92 | 94.07 | 87.83 | 88.28 | 88.45 |
| | FF/Female | 90.44 | 91.73 | 93.37 | 80.85 | 81.32 | 81.56 |
| | VGG/White | 92.68 | 91.15 | 90.39 | 80.28 | 80.72 | 80.8 |
| | VGG/Black | 97.9 | 97.74 | 98.74 | 87.74 | 88.94 | 89.25 |
| | VGG/Asian | 94.83 | 95.42 | 97.1 | 93.49 | 94.43 | 94.72 |

Table 14: White-box ASR(%) on $F_1$, "FF" indicates FairFace.

| $F_{test}$ | $F^{-1}$ | $\mathcal{D}_f$ | Target Dataset | | | | | |
|---|---|---|---|---|---|---|---|---|
| | | | LFW | CFP | AGE | FairFace | | |
| | | | | | | $\tau_{\text{LFW}}$ | $\tau_{\text{CFP}}$ | $\tau_{\text{AGE}}$ |
| $F_2$ | $F_{1_A}^{-1}$ | Fair/Male | 82.44 | 87.89 | 92.68 | 66.93 | 76 | 78.73 |
| | | Fair/Female | 77.14 | 83.96 | 90.95 | 64.66 | 73.43 | 75.47 |
| | | VGG/White | 98.59 | 98.67 | 99.92 | 47.14 | 63.25 | 67.83 |
| | | VGG/Black | 77.48 | 90.38 | 93.59 | 38.33 | 52.11 | 57.02 |
| | | VGG/Asian | 68.03 | 78.67 | 78.43 | 46.98 | 59.94 | 64.2 |
| | $F_{1_B}^{-1}$ | Fair/Male | 60.74 | 74.49 | 71.93 | 59.94 | 69.26 | 72.22 |
| | | Fair/Female | 53.76 | 70.83 | 77.86 | 52.75 | 61.96 | 64.73 |
| | | VGG/White | 87.68 | 88.29 | 90.14 | 32.73 | 48 | 52.62 |
| | | VGG/Black | 60.57 | 79.14 | 84.44 | 28.07 | 40.67 | 45.52 |
| | | VGG/Asian | 56.35 | 76.55 | 86.19 | 39.69 | 54.51 | 59.52 |
| $F_3$ | $F_{1_A}^{-1}$ | Fair/Male | 94.25 | 93.78 | 96.41 | 84.31 | 88.22 | 89.6 |
| | | Fair/Female | 89.21 | 88.45 | 92.31 | 78.28 | 83.67 | 85.43 |
| | | VGG/White | 99.68 | 99.34 | 99.94 | 80.69 | 87.9 | 90.01 |
| | | VGG/Black | 95.16 | 95.04 | 96.8 | 67.28 | 78.84 | 82.54 |
| | | VGG/Asian | 81.78 | 82.76 | 80.13 | 67.83 | 78.12 | 81.76 |
| | $F_{1_B}^{-1}$ | Fair/Male | 82.17 | 88.7 | 89.26 | 72.26 | 79.62 | 82.39 |
| | | Fair/Female | 75.92 | 85.06 | 90.51 | 62.15 | 70.17 | 73.12 |
| | | VGG/White | 91.68 | 90.57 | 90.36 | 57.2 | 69.16 | 73.15 |
| | | VGG/Black | 88.36 | 93.63 | 96.85 | 44.85 | 62.32 | 68.75 |
| | | VGG/Asian | 82.01 | 90.57 | 95.49 | 52.68 | 69.26 | 76.13 |

Table 15: White-Box Transfer Attack Success Rate(%), where $F_l : F_1$.

To further evaluate the transferability of the white-box attack, we performed a transfer attack; however, due to space limitations, the results are included in this section. The test models $F_{test}$ for the transfer

attack are $F_2$ and $F_3$, and the detailed settings for the attack are identical to those described in the main text. The results of the attack success rates are presented in Tab. 15. The results generally demonstrate high success rates. While the success rates are relatively lower compared to the black-box direct attack, which has more information about the test model, they still confirm that the attack retains sufficient transferability to the test models even in scenarios where no prior information is available.

## G.2  Effect of Correction Matrix $R$ in Black-box

As mentioned in the main text, we propose an additional technique, the correction matrix, to better refine the scores obtained in the black-box setting at the local level. The correction matrix essentially means adjusting the scores obtained from the target model to make them more suitable for local use, which can be thought of as helping the adversarial face to align with the same identity as the target image. However, since the ASR used in the main text excludes images with the intended attribute from the selection of target images and also considers whether the attack results reflect the intended attribute when determining the success of the attack, it may not be the most suitable metric for analyzing the effect of $R$. Therefore, to analyze the effect of $R$, we define the Identity Matching Rate (IMR) as follows:

$$\text{IMR} = \frac{\sum_{i=1}^{|\mathcal{I}|} \mathbb{1}(S_{test}(\text{img}_i, \widehat{\text{img}}_i) \geq \tau_{test})}{|\mathcal{I}|}$$

, where $\mathcal{I}$ represents the set of total images, $\text{img}_i \in \mathcal{I}$ refers to each individual target image, and $\mathbb{1}(\cdot)$ is a function mapping 1 if the input statement is a true 1, and 0 otherwise. $\widehat{\text{img}}_i$ is an adversarial face generated by an attack algorithm. $S_{target}(\text{Img}_1, \text{Img}_2)$ gives a cosine similarity score (resp. confidence score) between $\text{Img}_1$ and $\text{Img}_2$ from the open-source (resp. commercial) FRS. If the score exceeds the predefined threshold $\tau_{target}$ of the test FRS, $\text{Img}_1$ and $\text{Img}_2$ are considered as the same identity in the target FRS.

We performed the black-box attack with the same setup mentioned in the main text. We then compared the IMR before and after applying $R$ and marked the results in Tab. 16: if the IMR increased after applying $R$, it is highlighted in blue, and if it decreased, it is marked in red. As shown in the table, $R$ had a positive impact on identity matching in most cases. The results in this table are also reflected in the black-box direct attack success rate presented in Tab. 2 of the main text, with only the areas where the effect of $R$ has a negative impact highlighted in blue.

| $T$ | $F^{-1}$ | $D_f$ | Target Dataset | | | | | |
| | | | LFW | CFP | AGE | FairFace | | |
| | | | | | | $\tau_{\text{LFW}}$ | $\tau_{\text{CFP}}$ | $\tau_{\text{AGE}}$ |
| $F_2$ | $F_{1_A}^{-1}$ | Fair/Male | +1.20 | +0.17 | +0.11 | +1.25 | +0.32 | +0.12 |
| | | Fair/Female | +1.96 | +0.36 | +0.04 | +0.75 | +0.36 | +0.23 |
| | | VGG/White | +0.53 | +0.07 | +0.02 | +1.57 | +0.41 | +0.22 |
| | | VGG/Black | +0.09 | 0 | +0.01 | -0.06 | -0.07 | -0.06 |
| | | VGG/Asian | +0.75 | +0.20 | +0.04 | +0.54 | +0.14 | +0.17 |
| | $F_{1_B}^{-1}$ | Fair/Male | +0.88 | +0.67 | +0.67 | +2.40 | +1.39 | +1.02 |
| | | Fair/Female | +2.12 | +1.94 | +1.76 | +3.47 | +1.64 | +1.40 |
| | | VGG/White | +0.97 | +0.55 | +0.13 | +1.94 | +0.61 | +0.37 |
| | | VGG/Black | +0.77 | +0.46 | +0.26 | +1.64 | +1.00 | +0.74 |
| | | VGG/Asian | +3.60 | +2.07 | +1.73 | +2.81 | +1.78 | +1.22 |
| $F_3$ | $F_{1_A}^{-1}$ | Fair/Male | 0 | +0.03 | 0 | +0.03 | -0.02 | -0.02 |
| | | Fair/Female | +0.19 | -0.04 | +0.10 | -0.30 | +0.02 | +0.01 |
| | | VGG/White | +0.02 | 0 | 0 | +0.01 | +0.02 | +0.01 |
| | | VGG/Black | -0.06 | 0 | 0 | -0.05 | -0.02 | -0.01 |
| | | VGG/Asian | +0.01 | -0.14 | 0 | -0.15 | +0.01 | 0 |
| | $F_{1_B}^{-1}$ | Fair/Male | +1.46 | +0.73 | +0.49 | -0.41 | -0.44 | -0.34 |
| | | Fair/Female | +0.30 | +1.42 | +0.82 | +0.77 | +0.49 | +0.59 |
| | | VGG/White | 0 | +0.41 | +0.16 | +1.10 | +0.37 | +0.28 |
| | | VGG/Black | +1.80 | +0.28 | +0.50 | +0.73 | +0.73 | +0.57 |
| | | VGG/Asian | +2.79 | +2.06 | +1.31 | +3.55 | +1.66 | +0.81 |

Table 16: Change in IMR after applying correction matrix $R$: blue for increase, red for decrease.

In addition to the analysis of $R$, we provide a table to illustrate how well our proposed attack performs in terms of identity matching alone. The shared attack settings include the identity matching rates for the white-box direct attack and the black-box direct attack, with the detailed settings identical to those mentioned in the main text. The results are presented in Tab. 17 and Tab. 18, respectively, in sequential order. As shown in the numbers within the tables, our method achieves excellent IMRs.

| $F^{-1}$ | $D_f$ | Target Dataset | | | | | |
|---|---|---|---|---|---|---|---|
| | | LFW | CFP | AGE | FairFace | | |
| | | | | | $\tau_{\text{LFW}}$ | $\tau_{\text{CFP}}$ | $\tau_{\text{AGE}}$ |
| $F_{1_A}^{-1}$ | Fair/male | 99.95 | 100 | 100 | 99.71 | 99.93 | 99.98 |
| | Fair/female | 99.87 | 100 | 100 | 99.61 | 99.94 | 99.99 |
| | VGG/White | 99.99 | 99.98 | 100 | 99.62 | 99.87 | 99.97 |
| | VGG/Black | 99.97 | 100 | 100 | 99.55 | 99.86 | 99.95 |
| | VGG/Asian | 100 | 100 | 100 | 99.87 | 99.98 | 99.98 |
| $F_{1_B}^{-1}$ | Fair/male | 98.58 | 99.52 | 99.84 | 99.09 | 99.65 | 99.88 |
| | Fair/female | 96.08 | 99.03 | 99.88 | 98.96 | 99.63 | 99.86 |
| | VGG/White | 99.87 | 99.85 | 99.98 | 98.79 | 99.58 | 99.78 |
| | VGG/Black | 99.1 | 99.66 | 99.99 | 96.94 | 98.79 | 99.37 |
| | VGG/Asian | 98.91 | 99.6 | 99.99 | 98.6 | 99.57 | 99.86 |

Table 17: White-Box Direct Attack IMR(%).

| $T$ | $F^{-1}$ | $D_f$ | Target Dataset | | | | | |
|---|---|---|---|---|---|---|---|---|
| | | | LFW | CFP | AGE | FairFace | | |
| | | | | | | $\tau_{\text{LFW}}$ | $\tau_{\text{CFP}}$ | $\tau_{\text{AGE}}$ |
| $F_2$ | $F_{1_A}^{-1}$ | Fair/Male | 98.53 | 99.71 | 99.96 | 98.52 | 99.7 | 99.7 |
| | | Fair/Female | 96.69 | 99.59 | 99.82 | 97.64 | 99.42 | 99.64 |
| | | VGG/White | 99.94 | 99.96 | 100 | 98.94 | 99.68 | 99.75 |
| | | VGG/Black | 99.49 | 99.94 | 100 | 99.19 | 99.84 | 99.89 |
| | | VGG/Asian | 98.3 | 99.94 | 99.99 | 98.5 | 99.54 | 99.78 |
| | $F_{1_B}^{-1}$ | Fair/Male | 79.02 | 92.89 | 95.37 | 82.79 | 94.31 | 96.1 |
| | | Fair/Female | 63.67 | 84.8 | 90.69 | 77.01 | 90.6 | 93.6 |
| | | VGG/White | 87.39 | 94.74 | 98.54 | 76.1 | 91.6 | 94.23 |
| | | VGG/Black | 87.88 | 96.23 | 97.78 | 88.38 | 96.92 | 98.17 |
| | | VGG/Asian | 80.13 | 94.11 | 97.33 | 88.67 | 96.78 | 97.96 |
| $F_3$ | $F_{1_A}^{-1}$ | Fair/Male | 99.62 | 99.99 | 99.98 | 99.85 | 99.97 | 99.97 |
| | | Fair/Female | 98.55 | 99.75 | 100 | 99.4 | 99.97 | 100 |
| | | VGG/White | 100 | 100 | 100 | 99.92 | 100 | 100 |
| | | VGG/Black | 99.68 | 99.99 | 100 | 99.79 | 99.97 | 99.99 |
| | | VGG/Asian | 99.29 | 99.84 | 100 | 99.7 | 99.99 | 100 |
| | $F_{1_B}^{-1}$ | Fair/Male | 84.91 | 94.84 | 96.24 | 89.64 | 96.67 | 98.26 |
| | | Fair/Female | 65.93 | 83.33 | 93.4 | 81.22 | 93.16 | 96.34 |
| | | VGG/White | 94.24 | 97.7 | 99.51 | 91.77 | 97.75 | 99.01 |
| | | VGG/Black | 88.02 | 96 | 98 | 89.05 | 97.38 | 98.79 |
| | | VGG/Asian | 78.64 | 92.78 | 97.75 | 87.61 | 96.58 | 98.35 |

Table 18: Black-Box Direct Attack IMR(%).

### G.3 Black-box Attack against target FRS $F_{\text{T}}$

In Tab. 19, we provide the ASR on commercial FRS $F_{\text{T}}$ using gender-attributed $\mathcal{D}_f$. Similar to ASR on $F_{\text{A}}$ in Tab. 3, we note that the ASR related to the FairFace dataset is significantly larger than the transfer attack. However, since the decrease in ASR is obvious for the LFW dataset, we leave analyzing and improving this phenomenon as a future topic.

### G.4 Experimental Setup for Table 4 and Table 5

We base our comparison on [63], which follows the experimental protocol of CLIP2Protect [68]. Specifically, 500 subjects are selected from the CelebA-HQ dataset [36], each with a pair of facial images. In their setup, one image from each pair is used to generate adversarial examples (training set), and the other is used to evaluate the attack success rate (test set). In our comparison, we used the same

| $T$ | $F^{-1}$ | Target | $\mathcal{D}_f$ | with $R$ | | without $R$ | |
|---|---|---|---|---|---|---|---|
| | | | | $\tau = 0.8$ | $\tau = 0.99$ | $\tau = 0.8$ | $\tau = 0.99$ |
| | | | Transfer Attack without Queries (The images from Table 14 were used.) | | | | |
| $F_\mathsf{T}$ | $F_{1_A}^{-1}$ | LFW | VGG/White | N/A | | 91.56 | 61.78 |
| | | | VGG/Black | N/A | | 49.79 | 10.00 |
| | | | VGG/Asian | N/A | | 45.71 | 9.98 |
| | | FairFace | VGG/White | N/A | | 0 | 0 |
| | | | VGG/Black | N/A | | 0.02 | 0 |
| | | | VGG/Asian | N/A | | 0 | 0 |
| | | | Direct Attack with Score Queries | | | | |
| $F_\mathsf{T}$ | $F_{1_A}^{-1}$ | LFW | VGG/White | 81.78 | 25.33 | 27.11 | 2.22 |
| | | | VGG/Black | 75.26 | 10.74 | 20.84 | 2.84 |
| | | | VGG/Asian | 56.26 | 4.41 | 13.23 | 0 |
| | | FairFace | VGG/White | 72.87 | 12.90 | 17.55 | 0.66 |
| | | | VGG/Black | 58.97 | 13.24 | 7.55 | 0.44 |
| | | | VGG/Asian | 62.90 | 7.37 | 12.78 | 0.49 |

Table 19: Black-box ASR on $F_\mathsf{T}$ using local FR model $F_1$

500 subjects and adopted the same data split. For the method of [63], we used our three open-source face recognition models ($F_1$, $F_2$, $F_3$) as surrogate models to generate adversarial examples, consistent with the original protocol which involves multiple surrogate networks. However, our method differs fundamentally in that it does not require a source image. Instead, we project feature vectors into attribute-specific subspheres (e.g., gender, race) to generate adversarial faces. Therefore, for each training image, we created one adversarial face per attribute category and reported the attribute-wise transfer attack success rates against AWS CompareFace, as shown in Tab. 4.

To ensure a fair comparison, we also adapted the method of [63] to our scenario by randomly sampling 500 images from the "100k Faces Generated by AI" dataset [61] as source images. Each source image was targeted toward a corresponding identity from the training set, and the adversarial faces were evaluated by querying AWS CompareFace. Importantly, unlike [63], our method supports attacks directly through actual API queries. We thus separately report our query-based black-box attack

| Target Image | [63] | | Ours | |
|---|---|---|---|---|
| | Source | Result | Transfer | Direct |
|  |  |  |  |  |
| scores | 0.0028 | 0.0147 | 0.7835 | 0.9635 |
|  |  |  |  |  |
| scores | 0.0115 | 0.4349 | 0.5410 | 0.9755 |
|  |  |  |  |  |
| scores | 0.0052 | 0.0267 | 0.1366 | 0.9834 |

Table 20: Additional examples for Tab. 5 using male, black, asian attributed subspheres, respectively.

success rates in Tab. 4, in addition to the transfer-based results. Furthermore, the evaluation against the test set, following the full protocol of [63], is included in Tab. 21 for completeness.

| Method | [63] | Ours (Transfer attack without queries) | | | | | Ours (Attack with 100 queries against AWS) | | | | |
|---|---|---|---|---|---|---|---|---|---|---|---|
| $f$ | [61] | Male | Female | White | Black | Asian | Male | Female | White | Black | Asian |
| $\tau = 0.8$ | 29.86 | 23.80 | 33.20 | 62.80 | 22.20 | 35.60 | 94.20 | 92.00 | 98.20 | 99.20 | 98.80 |
| $\tau = 0.99$ | 3.41 | 1.20 | 1.60 | 6.60 | 1.40 | 1.80 | 14.60 | 8.60 | 41.00 | 40.80 | 24.60 |

Table 21: ASR of [63] and ours evaluated on CelebA-HQ dataset [36], evaluated against test set images under different thresholds

## G.5 Black-box Attack on Non-facial Target

We conducted our proposed attack in a black-box setting on non-facial images. Specifically, we used the CIFAR-10 and Flower-102 test datasets and 10,000 randomly generated pixel images as the attack targets. CIFAR-10 is a widely used benchmark dataset for image classification, consisting of 10 categories, such as airplanes, cars, and animals. On the other hand, Flower-102 contains 102 categories of flowers with varying levels of visual complexity. In our black-box attack framework, $F_1$ was employed as the local model,

$F_2$ and $F_3$ served as the target models. The inverse models used were both of $F_{1_A}^{-1}$ and $F_{1_B}^{-1}$. The ASR of the attack under the aforementioned settings was presented in Tab. 22. Since the target datasets we selected are not verification datasets, the data is analyzed based on the threshold values derived from the best accuracy of the $F_2$ model on LFW, CFP-FP, and AgeDB. As evidenced by the attack success rates in the table, it was confirmed that the attack remains effective even when the target images are non-facial, but ASR is relatively lower compared to when the target image is facial images. Therefore, there is room for improvement in terms of extension to a broader target model, such as image classification models, of attack or non-facial adversarial examples on commercial systems.

## G.6 Ablation Study about $F^{-1}$ in Black-box

In our black-box algorithm, the inverse model for the local face recognition system is used in two distinct stages: once during preprocessing and the other once during the execution of the algorithm. Therefore, we conducted an ablation study using $F_1$, a face recognition model with two inverse models, $F^{-1}1_A$ and $F^{-1}1_B$. The results of this study are shared in Tab. 26

## G.7 Transfer Attack in Black-box

We confirmed that even in our black-box setting, the attack retains sufficient transferability to other test models that are not the target model. When the target model was $F_2$, $F_3$ was selected as the test model, and conversely, when the target model was $F_3$, $F_2$ was used as the test model for the

| $T$ | $F^{-1}$ | Target Dataset | Threshold : $\tau$ | | |
|---|---|---|---|---|---|
| | | | $\tau_{\text{LFW}}$ | $\tau_{\text{CFP}}$ | $\tau_{\text{AGE}}$ |
| $F_2$ | $F_{1_A}^{-1}$ | CIFAR-10 | 20.1 | 39.84 | 45.78 |
| | | Flower-102 | 15.8 | 31.87 | 37.9 |
| | | Random | 74.91 | 79.97 | 80.2 |
| | $F_{1_B}^{-1}$ | CIFAR-10 | 16.4 | 39.55 | 49.76 |
| | | Flower-102 | 11.57 | 32.42 | 40.67 |
| | | Random | 61.22 | 91.16 | 94.82 |
| $F_3$ | $F_{1_A}^{-1}$ | CIFAR-10 | 51.73 | 66.77 | 70.8 |
| | | Flower-102 | 58.35 | 63.83 | 64.62 |
| | | Random | 76.9 | 76.9 | 76.9 |
| | $F_{1_B}^{-1}$ | CIFAR-10 | 32.92 | 63.85 | 76.33 |
| | | Flower-102 | 51.14 | 77.89 | 86.01 |
| | | Random | 92.47 | 99.19 | 99.7 |

Table 22: Attack Success Rate(%) of Black-box Direct Attack on Non-facial Target Images $D_f$ uses VGG/Black, and the ASR is calculated based on the $\tau$ values of $F_2$ corresponding to each column.

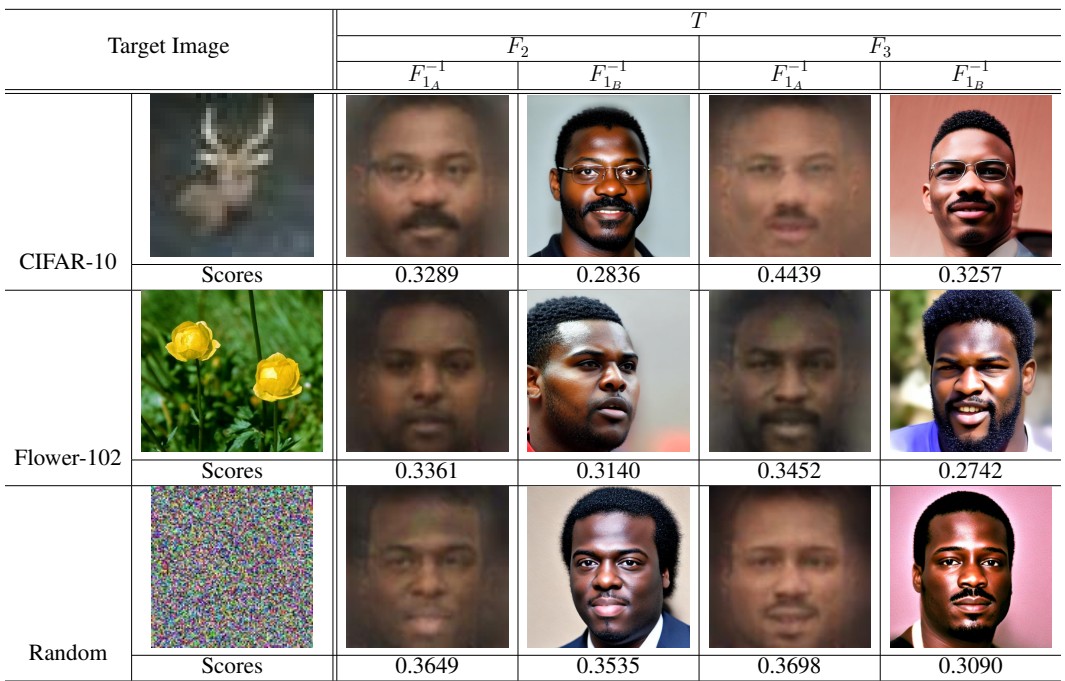

| Target Image | $T$ | | | |
| | $F_2$ | | $F_3$ | |
| | $F_{1_A}^{-1}$ | $F_{1_B}^{-1}$ | $F_{1_A}^{-1}$ | $F_{1_B}^{-1}$ |
| CIFAR-10 Scores | 0.3289 | 0.2836 | 0.4439 | 0.3257 |
| Flower-102 Scores | 0.3361 | 0.3140 | 0.3452 | 0.2742 |
| Random Scores | 0.3649 | 0.3535 | 0.3698 | 0.3090 |

Table 23: Black-box attack on non-facial images

attack. All other settings for the black-box attack are identical to those described in the main text. Additionally, since an ablation study on the use of inverse models, as discussed in Sec. G.6, is also feasible in this setting, we extracted the ASR results and included them in Tab. 27.

### G.8 Prototype of FRS for Strict Threshold

To construct a prototype FRS with a tight acceptance threshold, we designed a loss function with two key modifications to the original ArcFace [16] loss $L_A$. The loss function $L_A$ is defined as follows:

$$L_A(\mathbf{x}, W) = -\log \frac{e^{s\cos(\theta_{gt}+m)}}{e^{s\cos(\theta_{gt}+m)} + \sum_{j=1, j\neq gt}^{N_{id}} e^{s\cos\theta_j}},$$

where $gt$ is the index of the ground-truth, $m$ is the angular margin term, and $s$ is the scaling factor.

First, instead of applying the angular margin only to the target class as in the original ArcFace, we apply an inverted margin to non-target classes as well. Specifically, for the target class, we use the standard $\cos(\theta + m)$, while for all other classes, we use $\cos(\theta - m)$. This contrastive strategy enhances intra-class compactness by pulling feature vectors of the same identity closer together, while still maintaining sufficient inter-class separation.

Second, rather than computing cosine similarity over the entire 512-dimensional feature vector, we randomly split it into two 256-dimensional subspaces in each batch and calculate the cosine similarity separately in each subspace. This regularization strategy prevents the dominance of specific dimensions and promotes a more uniform distribution of information across all feature dimensions. As a result, the model learns to generate feature vectors that yield higher overall similarity between samples of the same identity. To formalize this idea, we define cosine similarity over each randomly selected subspace as follows. Let $x, y \in \mathbb{R}^d$. Let $I_1, I_2 \subseteq \{1, \ldots, d\}$ be two randomly sampled, disjoint subsets of indices such that $|I_1| = |I_2| = d/2$. Then, the partial cosine similarities is defined as follows:

$$\cos_1(x, y) := \frac{\langle x_{I_1}, y_{I_1}\rangle}{\|x_{I_1}\| \cdot \|y_{I_1}\|}, \quad \cos_2(x, y) := \frac{\langle x_{I_2}, y_{I_2}\rangle}{\|x_{I_2}\| \cdot \|y_{I_2}\|}$$

where $x_{I_1}$ and $y_{I_1}$ denote the subvectors of $x$ and $y$ corresponding to the index set $I_1$, respectively.

Then, for 1st (resp. 2nd) partial cosine similarities of $j$'th identity $\theta_j^{(1)}$ (resp. $\theta_j^{(2)}$), final loss function $L_P$ for our prototype FRS is defined as follows:

$$L_P(\mathbf{x}, W) = -\sum_{k=1}^{2} \log \frac{e^{s\cos(\theta_{gt}^{(k)}+m)}}{e^{s\cos(\theta_{gt}^{(k)}+m)} + \sum_{j=1, j\neq gt}^{N_{id}} e^{s\cos(\theta_j^{(k)}-m)}},$$

where $gt$ is the index of the ground-truth, $m$ is the angular margin term, and $s$ is the scaling factor.

These two modifications lead to significantly improved intra-class compactness, as shown in Fig. 14, where the decision threshold of the prototype FRS is formed around 0.3, which is higher than that of standard open-source models (typically around 0.2). This higher threshold increases the overall system's strictness against false accept. In Table 6, we report the ASR against the prototype FRS on the LFW dataset, alongside the results for other target models, $F_2$ and $F_3$.

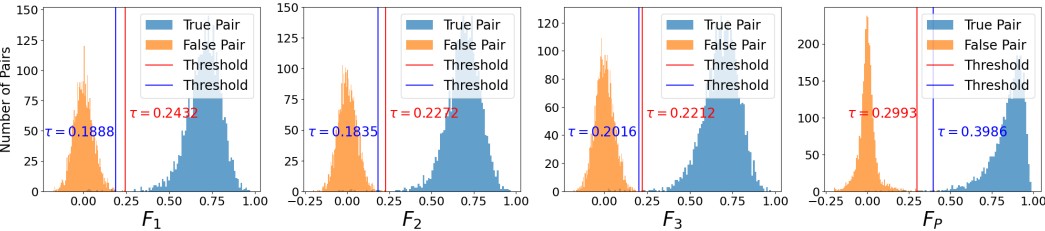

Figure 14: Cosine similarity histograms. The first three plots correspond to the open-source FRSs used in our experiments, whose thresholds are typically formed around 0.2. In contrast, our prototype FRS exhibits a significantly higher threshold, forming around 0.3 to 0.4. The blue line indicates the threshold at FAR = 0.1%, while the red line marks the threshold corresponding to the best accuracy.

### G.9 Adversarial Training and Certifiably Robust Face Recognition Models

In the literature of FR, several efforts have been made to achieve robustness against adversarial examples. Likewise to image classification, adversarial training [53] has been shown to be effective in defending against adversarial examples based on perturbations [80]. In addition, recently, Paik et al. [58] proposed a certifiably robust FR model against perturbation-based adversarial attacks. More precisely, they derived an upper bound of the magnitude of the adversarial perturbation that does not change the decision of the FR model. Since the proposed adversarial face generation method can be considered as adversarial examples, we conduct additional experiments to assess the robustness of the aforementioned (certifiably) robust FR models against our attack.

For adversarially trained FR models, we utilize the RobFR library that provides various pre-trained FR models with adversarial training, e.g., PGD-AT [53] and TRADES [86]. Among these models, we use two FR models using IResNet50 as the backbone and trained with CASIA-Webface [81] dataset, using ArcFace [16] and CosFace [74] loss functions, respectively. For the certifiably robust FR model, we used the pre-trained FR model provided by Paik et al. in their official implementation, which uses a custom 22-layer convolution neural network as a backbone, ElasticFace-Cos+ [3] as a loss function, and trained with the MS1MV3 [15] dataset. Detailed settings can be found in their original papers or their source codes.

We evaluate the ASRs of our attack against the above FR models under the same setting as in Sec. 4. In particular, for the comparison with the zeroth-order optimization (ZOO) based attack, we also provide the ASRs in the context of impersonation. More precisely, our implementation is based on Carlini & Wagner's white-box attack framework [6] with ADAM zeroth-order optimizer using a batch size of 128 with 20,000 iterations on one of the facial images. The goal is to make the image recognized as the same identity as a different person. We consider the attack successful if at least one perturbed image causes a successful impersonation.

The results are provided in Tab. 24. From this table, we can observe that, for the ZOO, all the adversarially trained FR models show low ASRs less than 4%, whereas the model from Paik et al. shows the ASR of at most 25%. On the other hand, for the proposed attack, we observe that the model by Paik et al. is vulnerable to our attack, whereas other adversarially trained models successfully defend against it. We hypothesize that such a dramatic difference is derived from two aspects: first, as

observed in the decision thresholds ($\tau$) of each target model, the threshold for adversarially trained models is substantially higher than those of Paik et al.'s FR model. In particular, the threshold for the latter is almost the same as that of non-robust FR models in the main text. From this, we can infer that the distribution of similarity scores from adversarially trained FR models is significantly far from that from the adversary's local model. Hence, the scores from queries would interrupt the interpolation over the attribute subsphere. On the other hand, when attacking Paik et al.'s FR model, Conj. 2 would be valid because of its similar decision threshold to that of the adversary's local model, thus succeeding the attack. However, it is important to note that, despite the success of the ZOO-based attack, the average of $L_2$ norm of these examples ranges up to 9, which is significantly exceeds the typical bounds of adversarial examples.

To further analyze our hypothesis, we also evaluated the ASRs from the transfer attack based on our attack, whose results are provided in Table 25. In particular, we considered two settings of the thresholds at FAR=0.1% and the best accuracy. From this table, we can observe that the transfer attack shows higher ASR than the black box setting for adversarially trained FR models, whereas the opposite tendency appears for Paik et al.'s model. This result supports our hypothesis discussed above; such a difference in ASRs indicates whether the adversary can make use of the queried scores for crafting an adversarial face or not.

Note that a direct comparison between ZOO and the our attack may be unfair because of the difference in the setting; for the former, the adversary adds a perturbation to one of the given pairs of images, i.e., the source image, whereas there is no such source image in the latter. Nevertheless, our attack reveals the potential vulnerability of these FR models, even in certifiably robust ones, in a practical setting. We leave the mitigation of our attack, along with an in-depth analysis on the relationship between our attack and prior perturbation-based attacks, as interesting yet important future work.

| Target Model | TAR(%) | $\tau$ | ZOO [7] | | Ours | | | | |
| --- | --- | --- | --- | --- | --- | --- | --- | --- | --- |
| | | | ASR (%) | Avg. $L_2$ | Male | Female | White | Black | Asian |
| PGD-Arc | 38.88 | 0.590 | 2.27 | 8.96 | 0 | 0 | 0 | 0 | 0 |
| PGD-Cos | 28.47 | 0.484 | 1.60 | 8.69 | 0 | 0 | 0 | 0 | 0 |
| Trades-Arc | 12.84 | 0.918 | 1.67 | 5.79 | 0 | 0 | 0 | 0 | 0 |
| Trades-Cos | 51.90 | 0.768 | 3.73 | 5.77 | 0 | 0 | 0 | 0 | 0 |
| Paik et al.[58] | 83.97 | 0.231 | 25.53 | 6.71 | 79.62 | 76.92 | 91.18 | 75.65 | 78.82 |

Table 24: ASR(%) against adversarially trained models [80] and certifiably robust model [58] on the LFW dataset. The FAR is set to 1e-3. Reconstructions are obtained using $F_{1_A}^{-1}$ as the inverse model.

| Target Model | TAR(%) | FAR(%) | $\tau$ | Ours | | | | |
| --- | --- | --- | --- | --- | --- | --- | --- | --- |
| | | | | Male | Female | White | Black | Asian |
| PGD-Arc | 38.88 | | 0.590 | 0.46 | 0.88 | 3.95 | 0.62 | 1.19 |
| PGD-Cos | 28.47 | | 0.484 | 0.15 | 0.47 | 1.19 | 0.55 | 0.57 |
| Trades-Arc | 12.84 | 0.1 | 0.918 | 2.58 | 2.09 | 12.20 | 0.88 | 1.54 |
| Trades-Cos | 51.90 | | 0.768 | 0.76 | 0.43 | 2.18 | 1.13 | 0.62 |
| Paik et al.[58] | 83.97 | | 0.231 | 14.96 | 15.31 | 39.34 | 11.60 | 15.11 |
| PGD-Arc | 84.23 | 7.23 | 0.357 | 9.26 | 13.94 | 39.65 | 17.06 | 19.99 |
| PGD-Cos | 84.03 | 9.67 | 0.260 | 6.91 | 7.20 | 22.36 | 10.54 | 10.48 |
| Trades-Arc | 59.53 | 16.63 | 0.785 | 63.02 | 58.55 | 82.55 | 47.10 | 50.85 |
| Trades-Cos | 88.46 | 8.67 | 0.678 | 6.61 | 5.32 | 19.66 | 11.23 | 4.14 |
| Paik et al.[58] | 94.80 | 3.2 | 0.165 | 35.31 | 39.50 | 67.31 | 31.71 | 37.20 |

Table 25: Transfer ASR(%) against adversarially trained models [80] and certifiably robust model [58] on the LFW dataset. The FARs are set by both its value at FAR = 1e-3, and by its value at the point of best accuracy, which is determined according to the LFW dataset evaluation protocol. Reconstructions are obtained using $F_{1_A}^{-1}$ as the inverse model.

| T | F$^{-1}$ Pre | F$^{-1}$ Alg | $D_f$ | LFW | CFP | AGE | FairFace $\tau_{\text{LFW}}$ | $\tau_{\text{CFP}}$ | $\tau_{\text{AGE}}$ |
|---|---|---|---|---|---|---|---|---|---|
| $F_2$ | $F_{1_A}^{-1}$ | $F_{1_A}^{-1}$ | Fair/Male | 97.29 | 99.41 | 99.18 | 97.25 | 96.5 | 97.93 |
| | | | Fair/Female | 93.33 | 96.02 | 97.44 | 93.26 | 94.16 | 93.75 |
| | | | VGG/White | 99.94 | 99.95 | 100 | 99.92 | 99.78 | 99.98 |
| | | | VGG/Black | 87.98 | 90.61 | 88.66 | 84.82 | 86.23 | 86.42 |
| | | | VGG/Asian | 73.82 | 73.09 | 72.27 | 68.48 | 68.07 | 67.45 |
| | | $F_{1_B}^{-1}$ | Fair/Male | 79.28 | 91.17 | 94.24 | 85.67 | 90.8 | 93.93 |
| | | | Fair/Female | 73.38 | 88.31 | 91.36 | 75.91 | 86.93 | 92.88 |
| | | | VGG/White | 95.31 | 96.59 | 97.06 | 95.89 | 95.48 | 96.55 |
| | | | VGG/Black | 91.16 | 96.16 | 96.94 | 91.52 | 96.07 | 97.03 |
| | | | VGG/Asian | 82.77 | 90.4 | 91.93 | 81.23 | 89.08 | 90.17 |
| | $F_{1_B}^{-1}$ | $F_{1_A}^{-1}$ | Fair/Male | 90.45 | 97.87 | 99.07 | 93.19 | 96.61 | 97.91 |
| | | | Fair/Female | 77.88 | 93.38 | 96.46 | 85.88 | 93.65 | 96.5 |
| | | | VGG/White | 95.98 | 98.92 | 99.88 | 99.46 | 99.69 | 99.91 |
| | | | VGG/Black | 85.42 | 89.9 | 87.96 | 85.25 | 86.38 | 85.17 |
| | | | VGG/Asian | 61.76 | 66.27 | 71.31 | 58.62 | 63.92 | 64.6 |
| | | $F_{1_B}^{-1}$ | Fair/Male | 70.82 | 86.66 | 91.16 | 75.59 | 88.14 | 90.53 |
| | | | Fair/Female | 58.48 | 80.25 | 86.2 | 60.09 | 78.5 | 88.32 |
| | | | VGG/White | 84.42 | 90.43 | 94.84 | 90.16 | 92.93 | 95.09 |
| | | | VGG/Black | 85.1 | 93.68 | 94.97 | 83.45 | 92.53 | 94.38 |
| | | | VGG/Asian | 73.97 | 84.9 | 89.81 | 70.66 | 81.57 | 86.4 |
| $F_3$ | $F_{1_A}^{-1}$ | $F_{1_A}^{-1}$ | Fair/Male | 98.28 | 99.28 | 99.42 | 97.29 | 97.44 | 97.44 |
| | | | Fair/Female | 94.84 | 96.63 | 96.75 | 95.65 | 96.24 | 96.27 |
| | | | VGG/White | 98.9 | 99.68 | 99.75 | 99.61 | 99.7 | 99.7 |
| | | | VGG/Black | 92.12 | 92.63 | 92.68 | 79.47 | 79.61 | 79.63 |
| | | | VGG/Asian | 78.91 | 79.53 | 79.63 | 76.99 | 77.22 | 77.22 |
| | | $F_{1_B}^{-1}$ | Fair/Male | 91.79 | 97.97 | 98.84 | 91.53 | 94.17 | 94.54 |
| | | | Fair/Female | 86.07 | 94.39 | 95.79 | 86.69 | 91.06 | 92.14 |
| | | | VGG/White | 89.58 | 97.25 | 98.16 | 89.27 | 90.71 | 90.97 |
| | | | VGG/Black | 87.99 | 91.51 | 91.87 | 90.65 | 93.73 | 94.14 |
| | | | VGG/Asian | 73.86 | 77.57 | 77.97 | 86.42 | 89.96 | 90.6 |
| | $F_{1_B}^{-1}$ | $F_{1_A}^{-1}$ | Fair/Male | 87.58 | 94.46 | 95.31 | 96.98 | 98.37 | 98.51 |
| | | | Fair/Female | 84.43 | 91.56 | 93.06 | 93.47 | 97.27 | 98.07 |
| | | | VGG/White | 90.28 | 95.92 | 96.63 | 97.94 | 99.24 | 99.32 |
| | | | VGG/Black | 91.75 | 96.34 | 96.78 | 82.38 | 83.98 | 84.13 |
| | | | VGG/Asian | 89.19 | 93.44 | 94.02 | 74.83 | 77.22 | 77.64 |
| | | $F_{1_B}^{-1}$ | Fair/Male | 81.31 | 91.94 | 93.63 | 85.76 | 93.1 | 94.65 |
| | | | Fair/Female | 75.21 | 88.31 | 91.02 | 77.09 | 88 | 90.81 |
| | | | VGG/White | 73.32 | 87.96 | 90.49 | 84.23 | 89.77 | 90.93 |
| | | | VGG/Black | 86.95 | 95.23 | 96.38 | 86.77 | 93.74 | 95.16 |
| | | | VGG/Asian | 83.69 | 90.95 | 91.98 | 82.61 | 90.01 | 91.56 |

Table 26: Ablation study ASR(%) for the use of inverse models during preprocessing and algorithm execution in the black-box setting.

| $T$ | $F_{test}$ | $F^{-1}$ | | $D_f$ | Target Dataset | | | | | |
| | | Pre | Alg | | LFW | CFP | AGE | FairFace | | |
| | | | | | | | | $\tau_{\text{LFW}}$ | $\tau_{\text{CFP}}$ | $\tau_{\text{AGE}}$ |
| $F_2$ | $F_3$ | $F_{1_A}^{-1}$ | $F_{1_A}^{-1}$ | Fair/Male | 64.57 | 78.92 | 81.48 | 28.03 | 42.02 | 49.5 |
| | | | | Fair/Female | 56.73 | 74.93 | 89.59 | 24.69 | 38.5 | 46.13 |
| | | | | VGG/White | 95.54 | 96.58 | 99.7 | 17.88 | 32.24 | 40.67 |
| | | | | VGG/Black | 59.74 | 77.61 | 77.96 | 13.95 | 23.81 | 29.93 |
| | | | | VGG/Asian | 45.37 | 58.77 | 66.81 | 13.51 | 23.14 | 29 |
| | | | $F_{1_B}^{-1}$ | Fair/Male | 37.08 | 50.21 | 53.07 | 15.89 | 26.66 | 33.18 |
| | | | | Fair/Female | 27.13 | 46.12 | 63.99 | 12.95 | 23.24 | 30.02 |
| | | | | VGG/White | 83.87 | 86.65 | 95.3 | 9.07 | 20.49 | 27.52 |
| | | | | VGG/Black | 42.01 | 63.19 | 66.83 | 6.5 | 13.6 | 18.89 |
| | | | | VGG/Asian | 30.1 | 51.15 | 68.85 | 7.7 | 15.56 | 21.6 |
| | | $F_{1_B}^{-1}$ | $F_{1_A}^{-1}$ | Fair/Male | 49.87 | 66.72 | 69.04 | 18.62 | 31 | 38.84 |
| | | | | Fair/Female | 34.71 | 57.85 | 72.93 | 14.87 | 26.92 | 34.41 |
| | | | | VGG/White | 83.07 | 87.76 | 98.59 | 9.61 | 21.67 | 29.83 |
| | | | | VGG/Black | 45.85 | 67.53 | 70.1 | 9.13 | 17.62 | 23.11 |
| | | | | VGG/Asian | 26.27 | 42.17 | 57.17 | 8.4 | 16.71 | 21.94 |
| | | | $F_{1_B}^{-1}$ | Fair/Male | 25.57 | 37.06 | 41.4 | 10.02 | 18.38 | 24.33 |
| | | | | Fair/Female | 16.47 | 31.96 | 46.47 | 7.99 | 16.37 | 22.01 |
| | | | | VGG/White | 57.34 | 66.49 | 86.26 | 4.14 | 10.61 | 15.9 |
| | | | | VGG/Black | 27.9 | 49.82 | 56.37 | 4.04 | 9.54 | 13.55 |
| | | | | VGG/Asian | 19.88 | 37.53 | 54.4 | 5.53 | 12.1 | 17.25 |
| $F_3$ | $F_3$ | $F_{1_A}^{-1}$ | $F_{1_A}^{-1}$ | Fair/Male | 66.64 | 81.08 | 80.19 | 26.97 | 39.98 | 44.61 |
| | | | | Fair/Female | 58.75 | 77.89 | 84.85 | 22.7 | 36.12 | 40.68 |
| | | | | VGG/White | 95.44 | 96.69 | 99.61 | 13.2 | 25.25 | 30.45 |
| | | | | VGG/Black | 57.04 | 73.79 | 74.4 | 11.32 | 18.98 | 22.09 |
| | | | | VGG/Asian | 43.13 | 57.56 | 62.14 | 11.1 | 20.33 | 23.92 |
| | | | $F_{1_B}^{-1}$ | Fair/Male | 37.68 | 50.97 | 54.44 | 19.47 | 31.58 | 35.88 |
| | | | | Fair/Female | 33.75 | 53.53 | 64.54 | 18.34 | 29.92 | 34.63 |
| | | | | VGG/White | 84.05 | 84.83 | 93.92 | 8.64 | 18.74 | 22.9 |
| | | | | VGG/Black | 43.55 | 64.9 | 67.75 | 8.28 | 15.57 | 18.93 |
| | | | | VGG/Asian | 35.44 | 56.99 | 68.42 | 9.7 | 18.79 | 22.47 |
| | | $F_{1_B}^{-1}$ | $F_{1_A}^{-1}$ | Fair/Male | 53.25 | 68.24 | 68.97 | 17.73 | 29.06 | 33.61 |
| | | | | Fair/Female | 39.36 | 62.68 | 73.98 | 13.42 | 23.29 | 27.43 |
| | | | | VGG/White | 85.44 | 88.88 | 98.88 | 7.63 | 16.87 | 21.41 |
| | | | | VGG/Black | 44.04 | 64.78 | 67.27 | 7.54 | 14.17 | 17.1 |
| | | | | VGG/Asian | 26.62 | 42.57 | 50.38 | 7.21 | 14.6 | 17.45 |
| | | | $F_{1_B}^{-1}$ | Fair/Male | 27.72 | 40.31 | 40.29 | 11.49 | 21.74 | 25.86 |
| | | | | Fair/Female | 19.79 | 36.45 | 47.89 | 10.53 | 19.54 | 23.2 |
| | | | | VGG/White | 63.13 | 70.14 | 86.12 | 4.72 | 11.67 | 15.14 |
| | | | | VGG/Black | 30.35 | 50.27 | 56.11 | 5.58 | 11.58 | 14.73 |
| | | | | VGG/Asian | 23.06 | 42.5 | 52.56 | 7.2 | 14.97 | 18.43 |

Table 27: Black-Box Transfer ASR(%).

