# OpenReview forum: "Non-Adaptive Adversarial Face Generation"
_NeurIPS.cc/2025/Conference — NeurIPS 2025 poster_

### Official Review · Reviewer_VhZn · 2025-06-30

**Clarity:** 4
**Significance:** 3
**Originality:** 4
**Rating:** 6
**Confidence:** 4

**Summary:**

The paper proposes a novel non-adaptive adversarial face generation method that does not rely on iterative optimization or surrogate models. It exploits the structure of the feature space in deep learning-based face recognition systems by assuming that individuals with similar attributes lie on an attributed subsphere. The method achieves state-of-the-art attack success rates.

**Questions:**

1. Are all facial attributes projectable into the subsphere?
2. See Weaknesses.

**Ethical Concerns:**

["NO or VERY MINOR ethics concerns only"]

**Final Justification:**

I will maintain my original score after considering the authors' response.

**Limitations:**

Yes.

**Paper Formatting Concerns:**

None.

**Quality:**

4

**Strengths And Weaknesses:**

### Strengths
1. The authors proposes a non-adaptive, non-iterative attack method based on the feature metric space of FRS.
2. The experiments are Comprehensive.
3. The paper is theoretically well-grounded, with rigorous and comprehensive derivations and proofs provided for each conjecture.

### Weaknesses
1. The experiments mainly consider gender and race, with limited discussion on other aspects such as facial expressions.
2. The generated images appear overly smoothed, which may increase the risk of them being identified as adversarial examples.

---

> ### Author Rebuttal · Authors · 2025-07-29
>
> ### **Response to Question & Weakness 1:**
>
> Thank you for your valuable comments. Although we primarily focus on gender and race attributes, we also visualize attribute-specific principal components for other annotations, such as age, eyeglasses, hat, and baldness in Appendix D (Figures 9 and 10). As discussed in Appendix D, we interpret that attributes that are critical for distinguishing identities are more likely to form an attribute subsphere. This is because face recognition models are designed to extract features that are discriminative for identity, while discarding irrelevant information such as pose, lighting, or background.
>
> To further examine whether other facial attributes form meaningful structures in the feature space, we conduct an additional analysis using the LFW dataset. Specifically, for each binary attribute in CelebA [a], we construct two PCA-based subspaces: one from CelebA samples *with* the attribute (positive set), and the other from samples *without* the attribute (negative set). Then, we project each LFW sample onto both subspaces and compute the cosine similarity between the two projected feature vectors. A *low cosine similarity* indicates that the two subspaces produce significantly different projections for the same sample, suggesting that the attribute introduces geometric separation and may form a coherent subspace. In contrast, a *high similarity* suggests that the attribute has little effect on the projection and is less likely to form a distinct subspace.
>
> This analysis reveals that identity-relevant attributes such as 'Male', 'Bald', and 'Gray_Hair' tend to produce low similarity scores, whereas high-entropy or transient attributes such as 'Smiling' or 'Mouth_Slightly_Open' result in higher similarity scores. While some attributes have highly imbalanced or small sample sizes (e.g., 'Rosy_Cheeks', 'Blurry', 'Wearing_Necklace'), which may limit the reliability of cosine similarity interpretation, others such as 'Male', 'Smiling', and 'Mouth_Slightly_Open' are well-balanced and occur frequently in the dataset. These cases are particularly informative: 'Male', which is nearly evenly distributed and identity-relevant, exhibits low cosine similarity, whereas 'Smiling' and 'Mouth_Slightly_Open', which are also frequent but temporary in nature, result in high similarity. This observation reinforces our hypothesis that identity-discriminative attributes are more likely to form structured subspheres, while expression-related attributes are not.
>
>
> | Attribute            | Cosine Similarity | Number of Belonging Samples | Number of Non-belonging Samples |
> |:----------------------|:-------------:|:-------------:|:-----------------:|
> | Bald                | 0.4221 | 4,547 | 198,052 |
> | **Male**                | 0.4522 | 84,434 | 118,165 |
> | Gray_Hair           | 0.4530 | 8,499 | 194,100 |
> | Chubby              | 0.4932 | 11,663 | 190,936 |
> | Double_Chin         | 0.4948 | 9,459 | 193,140 |
> | Mustache            | 0.4948 | 8,417 | 194,182 |
> | Eyeglasses          | 0.5053 | 13,193 | 189,406 |
> | No_Beard            | 0.5117 | 169,158 | 33,441 |
> | Goatee              | 0.5142 | 12,716 | 189,883 |
> | Sideburns           | 0.5226 | 11,449 | 191,150 |
> | Wearing_Necktie     | 0.5380 | 14,732 | 187,867 |
> | Wearing_Lipstick    | 0.5423 | 95,715 | 106,884 |
> | 5_o_Clock_Shadow    | 0.5435 | 22,516 | 180,083 |
> | Young               | 0.5942 | 156,734 | 45,865 |
> | Wearing_Hat         | 0.6123 | 9,818 | 192,781 |
> | Heavy_Makeup        | 0.6152 | 78,390 | 124,209 |
> | Receding_Hairline   | 0.6302 | 16,163 | 186,436 |
> | Big_Nose            | 0.6475 | 47,516 | 155,083 |
> | Bushy_Eyebrows      | 0.6500 | 28,803 | 173,796 |
> | Blond_Hair          | 0.6998 | 29,983 | 172,616 |
> | Bags_Under_Eyes     | 0.7075 | 41,446 | 161,153 |
> | Black_Hair          | 0.7140 | 48,472 | 154,127 |
> | Attractive          | 0.7164 | 103,833 | 98,766 |
> | Bangs               | 0.7242 | 30,709 | 171,890 |
> | Arched_Eyebrows     | 0.7439 | 54,090 | 148,509 |
> | Rosy_Cheeks         | 0.7525 | 13,315 | 189,284 |
> | Wearing_Earrings    | 0.7636 | 38,276 | 164,323 |
> | High_Cheekbones     | 0.7658 | 92,189 | 110,410 |
> | **Smiling**             | 0.7974 | 97,669 | 104,930 |
> | Wavy_Hair           | 0.7986 | 64,744 | 137,855 |
> | Wearing_Necklace    | 0.8066 | 24,913 | 177,686 |
> | Big_Lips            | 0.8151 | 48,785 | 153,814 |
> | Blurry              | 0.8195 | 10,312 | 192,287 |
> | Pointy_Nose         | 0.8279 | 56,210 | 146,389 |
> | Pale_Skin           | 0.8303 | 8,701 | 193,898 |
> | **Mouth_Slightly_Open** | 0.8458 | 97,942 | 104,657 |
> | Straight_Hair       | 0.8554 | 42,222 | 160,377 |
> | Brown_Hair          | 0.8616 | 41,572 | 161,027 |
> | Narrow_Eyes         | 0.8684 | 23,329 | 179,270 |
> | Oval_Face           | 0.8714 | 57,567 | 145,032 |
>
> **Table F: Cosine Similarity Between Attribute-Positive and Attribute-Negative Subspace Projections (LFW Samples)**
>
> Notably, a contemporaneous work [d] that appeared during the review period (July 15, 2025) provides supporting evidence of the above argument. In this work, they studied the geometrical structure of face features associated with various facial attributes at different levels of granularity. In particular, they quantified the discriminative power of each facial attribute for identity recognition by measuring the entropy of that attribute within an identity. An attribute with a high entropy tends to easily change within the same identity, so it would provide limited information for extracting discriminative facial features. From this, they found that attributes like gender and skin tone have low entropy, while expressions and pose exhibit high entropy. We believe that our cosine similarity analysis complements this view and provides empirical evidence supporting our subsphere hypothesis. We also consider exploring this direction more formally in future work.
>
> ### **Response to Weakness 2:**
>
> Thank you for pointing out this issue. As mentioned by the reviewer, because adversarial faces from our attack are outputs of a generative inverse model (e.g., Arc2Face, which is based on fine-tuned Stable Diffusion). Consequently, they may be susceptible to detection by deepfake or anti-spoofing classifiers. We fully acknowledge this limitation and note that adversarial robustness against detection is an orthogonal and important research question.
>
> That said, we emphasize that our attack pipeline—particularly the generation of projected features using attribute subspaces—is independent of the inverse model architecture. As demonstrated in Figure 7, other inverse models like NbNet can also leverage an attribute subsphere by interpolating its principal components. Hence, once a better inverse model that returns more naturalistic faces than Arc2Face, we can instead employ it in our attack pipeline in a plug-and-play manner. In addition, we also note that subsequent post-processings on the output adversarial faces—applying photographic filters or effectors—could be employed to bypass the detection methods. We will incorporate the above discussion into the final version to clarify potential attack vectors and defenses.
>
> [d] Leroy, Pierrick, et al. "Attributes Shape the Embedding Space of Face Recognition Models." arXiv preprint arXiv:2507.11372 (2025); To appear ICML 2025.

---

> > ### Comment · Reviewer_VhZn · 2025-08-04
> >
> > Thank you for the rebuttal. I will maintain my original score after considering the authors' response.

---

### Official Review · Reviewer_2Myn · 2025-07-01

**Clarity:** 3
**Significance:** 3
**Originality:** 3
**Rating:** 6
**Confidence:** 3

**Summary:**

This paper first investigated the feature subspaces of different attributes in the metric space of facial recognition systems (FRS). Based on the investigation, the authors develop a process to explore the feature space with universal principal components and extend it to a non-adaptive adversarial attack algorithm for FRS. Experiment results across various benchmarks and open-world systems demonstrate the effectiveness of the proposed adversarial attack algorithm.

**Questions:**

Have the authors consider combining the adversarial iterative techniques with the principal component methods?

**Ethical Concerns:**

["Major Concern: Improper research involving human subjects"]

**Final Justification:**

The authors' rebuttal has addressed my questions, so I have decided to raise my score to 6.

**Limitations:**

Yes.

**Quality:**

3

**Strengths And Weaknesses:**

### Strengths

1. **[New Insights].** Overall, the idea of leveraging principal components to generate adversarial samples for facial recognition systems is novel, as the authors provide new insights of leveraging the feature subspace of the metric learning model for adversarial attacks.

2. **[Theoretical Foundation].** The provided proof for Proposition 1 (expected projection distance) and empirically validates Conjectures 1–2 (existence of attributed subspheres and universal bases). The link between PCA-based subspheres and adversarial efficacy is insightful.

3. **[Paper Clarity].** The paper is overall well-written and easy to follow.

### Weaknesses

1. **[Comparisons with Iterative Adversarial Methods].** Despite achieving good performances by the proposed method, it seems that the authors didn't compare them with previous iterative-based adversarial ones. Comparisons across computational time and performances should be included.

---

> ### Author Rebuttal · Authors · 2025-07-29
>
> ### **Response to Weakness (Comparison with Iterative Adversarial Methods):**
>
> Thank you for your valuable question regarding comparisons with iterative adversarial methods.
>
> First, we would like to clarify the fundamental distinction between our method and existing iterative approaches. Prior works can largely be categorized into two groups: (1) traditional iterative attacks such as PGD, and (2) optimization-based adversarial face generation techniques that utilize GANs [81, 27, 67, 44] or diffusion models [58]. While these methods achieve impressive performance, they rely on iterative solvers and gradient computation, inherently assuming a white-box setting. More importantly, when attacking a black-box system, they must fall back on transferability-based strategies, which tend to result in lower attack success rates. Note that, as seen in Table 3 (our method) and Table 12 (PGD), the attack success rate drops drastically when targeting the FairFace dataset. This highlights a limitation of transferability-based attacks in face recognition: when the feature distributions of the local and target models are misaligned, the effectiveness of such attacks diminishes significantly.
>
> Despite this fundamental difference, we adopted the publicly available implementation of the recent diffusion-based approach from [58] (CVPR 2025) as a representative iterative method of (2) for empirical comparison. This method assumes access to a source image and incrementally edits it to match a target identity, in contrast to our method which does not rely on any source image or identity. While [58] offers flexible control over identity impersonation from a given source, our framework does not currently support this level of instance-level conditioning. This difference highlights an important limitation of our method: it focuses on generating adversarial faces with a desired source attribute rather than reproducing a specific source identity or source image. We believe that extending our framework toward source-identity-conditioned generation is not only a meaningful next step, but also an inherently challenging one—especially given our conjecture that the identity-conditioned subsphere may be fundamentally much narrower than attribute-conditioned subspheres, making adversarial projection may be infeasible.
>
> We include a direct comparison with [58] in Table 4 (attack success rates), Table 5 (qualitative examples), and Appendix G.4 (experimental setup). When evaluated under a transfer attack setting, our method achieves comparable performance to [58]. However, once black-box query access is leveraged, our approach significantly outperforms [58] in terms of attack success rate—despite requiring only a single set of 100 queries to the black-box target system, thanks to its non-adaptive, one-shot nature.
>
> Regarding traditional iterative attack methods (1) such as PGD, we note that while these methods achieve high success in white-box settings, their effectiveness in real-world black-box scenarios is limited due to their reliance on transferability. Nevertheless, to provide an estimate of the performance gap, we plan to include PGD-based white-box attacks transferred to commercial systems (e.g., AWS) in the final version.
>
> Additionally, we have implemented Zeroth-Order Optimization (ZOO)-based adversarial attack [6], which operates under a truly black-box setting using only score outputs—similar to ours. This method originally proposed in the image classification domain, but it is one of the similar attack to ours in terms of assumptions. We adapted ZOO to face recognition model and summarize the comparison below:
>
>
> | Method        | Setting             | Average Queries (successful cases only) | Max Queries (Batch × Iteration) | Attack Success Rate on $F_2$ | Attribute       | Avg. L2 Norm |
> |---------------|---------------------|------------------------|------------------------------|----------------------|------------------|---------------|
> | ZOO [6]       | Black-box (iterative)| 28,068                 | 40,000 ( 2 × 20,000)      | 44.33%               | N/A              | 6.186         |
> | Ours          | Black-box (1-shot)   | 100                        | 100 ( 100 × 1)            | 97.29%, 93.33%       | Male, Female     | N/A           |
>
> **Table D: Evaluating Query Efficiency and Attack Success Rates Under the Same Black-box Setting using LFW Dataset**
>
> While the success rate of ZOO approaches 50%, the method demands an enormous number of queries per image—typically exceeding 10,000, which is a hundread times more than ours—each with imperceptibly small perturbations. Such adaptive queries may be easily detected by query-based defense mechanisms in commercial APIs. In contrast, our method remains query-efficient, stealthy, and capable of generating attribute-specific adversarial faces in a single step.
>
> Given its relevance, we promise to including ZOO in the final version as an important baseline under the same black-box constraint.
>
> ### **Response to Question (Combining the Adversarial Iterative Technique with the Ours):**
>
> Thank you for the insightful question. While our current framework is non-adaptive and one-shot by design, we agree that integrating our approach with iterative adversarial methods is a compelling direction.
>
> In particular, one promising strategy is to first use our method to generate an adversarial face that exhibits the desired source attribute (e.g., opposite gender), and then apply a conventional iterative method (e.g., PGD or diffusion-based attack) to further push this image toward a target identity in feature space of local model. Because our attribute-specific subspace projection is semantically robust and empirically shown to generate feature vectors that are already close to the target identity in the feature space, the resulting image may already lie near the decision boundary. In this sense, our output can be viewed as a strong initialization for an iterative attack process, potentially improving the attack success rate.
>
> Moreover, this hybrid approach would still retain semantic control over the source attribute, while leveraging the optimization power of iterative attacks—offering the best of both sides. Notably, the adversarial face generated by our method does not correspond to any real identity in the training data; it is a synthetic construction optimized to possess the desired attribute while being deceptively close to the target identity in feature space. This makes it a compelling initialization for further optimization in iterative attacks.
>
> However, it remains an open question how best to combine our attribute-specific projection method with existing iterative optimization techniques. Such hybrid attacks may face non-trivial challenges: the compatibility between the projected feature space and iterative objectives is not guaranteed; the effectiveness may vary depending on the attribute in question; and gradient-based optimization could potentially degrade quality of attribute. Thus, although this combination is promising in theory, it requires further investigation.
>
> As a preliminary study, we implemented a prototype of hybrid attack by initializing a PGD-style iterative optimization using adversarial images generated from our method. The table below shows attack success rates under various threshold conditions ($\tau=0.8$, $\tau=0.99$), comparing cases with and without the additional PGD iterative solver using a small noise budget $\epsilon=0.25$. This experiment was conducted against the same commercial black-box system (AWS Rekognition API) used in our main experiments using both LFW and FairFace datasets across Male and Female attributes.
>
> | Dataset    | Attribute | Ours ($\tau=0.8$, Table 3) | Ours+PGD ($\tau=0.8$) | Ours ($\tau=0.99$, Table 3) | Ours+PGD ($\tau=0.99$) |
> |------------|-----------|----------------|------------------|-------------------|---------------------|
> | LFW        | Male      | 91.45%     | 91.76% (+0.31%)    | 5.95%         | 6.74% (+0.79%)       |
> | LFW        | Female    | 86.46%    | 86.46% (same)       | 4.51%         | 5.47% (+0.96%)       |
> | FairFace   | Male      | 93.87%     |  95.17% (+1.30%)   | 13.70%        | 21.19% (+7.49%)     |
> | FairFace   | Female    | 91.00%    | 93.38% (+2.38%)    | 12.33%        | 19.03% (+6.7%)      |
>
> **Table E: Prototype Hybrid Attack Results with No Extra Queries on AWS Rekognition API**
>
> Note that the PGD iterative solver operates on the attacker's local model, refining the adversarial image to better match the target identity in the semantic space, while remaining close in appearance to the initial adversarial face reconstructed by our proposed attack. Crucially, the total number of queries to the black-box system remains unchanged—only a single set of queries is performed prior to PGD iterative solver, thus preserving the query-efficiency of our original approach. While retaining the key advantage of our attack, this refinement yields up to a 7.49% increase in attack success rates, demonstrating the potential of this hybrid strategy.
>
> Additionally, we consider a more flexible setting by relaxing the original assumptions of our attack, such as the use of non-adaptive queries and strict query budget constraints to the black-box model. Under this relaxed setting, and by leveraging techniques such as ZOO, we believe that combining our approach with iterative solver-based attacks can serve as an effective initialization strategy for achieving higher identity similarity on the black-box target model.
>
> We will include the results of this prototype hybrid attack in the final version of our manuscript.

---

> > ### Comment · Reviewer_2Myn · 2025-08-05
> >
> > Thank you for your rebuttal. I'm satisfied with the response and thereby raise my score to 6.

---

### Official Review · Reviewer_phZN · 2025-07-01

**Clarity:** 2
**Significance:** 3
**Originality:** 3
**Rating:** 4
**Confidence:** 5

**Summary:**

This paper presents a method for generating adversarial facial images that can impersonate a target identity in face recognition systems (FRSs), even under black-box constraints. Unlike existing approaches that rely on iterative optimization or surrogate models, the proposed approach exploits the geometric structure of FRS feature spaces. Based on empirical observations that individuals with similar attributes (e.g., race or gender) cluster into specific subspaces, the paper uses projections onto these subspaces to generate adversarial images efficiently. The proposed method achieves over 93% attack success rate ono commercial APIs like AWS CompareFaces at default thresholds.

**Questions:**

Are there other subspace structures that would be more robust? Have the authors considered sparse representations such as those discussed in the 2009 PAMI paper by Wright et al. ? The authors point to the shortcomings of GAN-based methods. Have the authors considered the effectiveness of diffusion models?

**Ethical Concerns:**

["NO or VERY MINOR ethics concerns only"]

**Final Justification:**

I read the response from the reviewers and will keep the original score of boderline accept.

**Limitations:**

Section 5.3 discusses the limitations of the proposed method.

**Paper Formatting Concerns:**

None.

**Quality:**

3

**Strengths And Weaknesses:**

Strengths:
The attack remains effective even without access to gradients or the internals of the FRS, marking it as non-adaptive and non-iterative. Impressive attack success rate on commercial APIs and dating websites even under black-box attack scenarios.
Weaknesses: Using PCA dates the work back to the nineties. PCA representations are sentitive to illumination changes, rotation etc. What effcet will these have on attack performance? The paper is written in a style that makes it very difficult to comprehend. Too much notation for describing obvious concepts.

---

> ### Author Rebuttal · Authors · 2025-07-28
>
> ### **Response to Weakness, Question 2 (Comparison with Classical PCA and Wright et al.):**
>
> We appreciate the reviewer’s concern regarding the limitations of traditional pixel-level PCA. However, our method performs PCA on high-level deep feature vectors extracted by a deep learning-based face recognition model (e.g., ArcFace), rather than raw pixel intensities. Unlike classical approaches such as Eigenface that operate on pixel-level grayscale images and are highly sensitive to low-level variations (e.g., illumination, pose), our approach leverages deep features that are trained via metric learning to encode identity-relevant semantics while suppressing irrelevant variations. Such variations—like alignment and rotation—are commonly introduced during training as data augmentations, making the resulting feature vectors robust to them. This enables our PCA to operate on a semantically meaningful space shaped primarily by identity-related information.
>
> Moreover, while classical PCA-based methods like Eigenface aim to enhance face recognition accuracy by reducing dimensionality and improving class separability in the pixel domain, our use of PCA serves a fundamentally different purpose. We apply PCA not for classification but to uncover the geometric structure of face feature vectors associated with specific attributes in the deep feature space. This allows us to construct semantically meaningful subspaces (or “attributed subspheres”) aligned with the intrinsic geometry of the recognition model’s feature space, as evidenced by our interpolations along principal components in Figures 3, 7, and 11, where reconstructed images via inverse models consistently reflect the target attributes.
>
> In contrast to our approach, Sparse Representation-based Classification (SRC) from [b], as noted by the reviewer, addresses some of Eigenface’s limitations by expressing input images as sparse linear combinations of training samples. While SRC improves robustness to occlusion and noise, it still operates in the pixel domain and relies on solving sparse coding problems over exemplar dictionaries, which differs significantly from our feature-space approach. By performing PCA on deep feature vectors and combining it with well-trained inverse models like NbNet [50] or Arc2Face [54], our method generates noise-free adversarial examples in the pixel domain that preserve both identity and the desired target attribute, all within a non-adaptive, one-shot attack process.
>
> To further clarify the conceptual distinction between our method and pixel-based approaches, we summarize the differences in the following table:
>
> | Method | Phase | Domain | Input  | Output  | Purpose  |
> | -- | -- | -- | -- | -- | -- |
> | **Classical PCA**   | Subspace Construction   | Pixel | Grayscale face images | Principal components (Eigenfaces) | Capture appearance variations for dimensionality reduction  |
> |  | Application  | Pixel  | Grayscale face image  | Low-dimensional projection | Face recognition or image reconstruction  |
> | **SRC [b]** | Dictionary Construction | Pixel | Grayscale face images | Dictionary of exemplar images       | Build sparse representation basis for classification |
> |  | Application | Pixel | Grayscale face image | Sparse coefficient vector | Identify class using sparse coefficient vector |
> | **Ours**  | Subspace Construction   | Feature  | Deep features annotated by attribute | Attribute-specific PCA basis        | Generate both the attribute-specific projection matrix (PCA basis) and the corresponding reconstructed facial images for black-box system queries |
> |  | Black-box Application   | Score → Feature | Scores from queries against target system  | Feature vector |Infer latent representation from the target black-box model instead of a local model  |
> |  | Application  | Feature → Pixel | Feature vector of image  | Adversarial image via inverse model | Non-adaptive attack with semantic control (attribute-preserving) |
>
> **Table B: Comparison of Classical, SRC [b], and Our Method Across Construction and Application Stages**
>
> ### **Response to Question 1 (More Robust Subspace):**
>
> To further investigate the robustness of our attribute-specific subspaces, we conducted a white-box experiment analyzing the semantic stability of projected features across subspaces constructed under varying conditions. Specifically, we used the VGGFace dataset to construct gender-specific subspaces (e.g., Subspace_Male$_1$ and Subspace_Male$_2$) from disjoint subsets of feature vectors (e.g., Male$_1$ and Male$_2$), and varied both the number of samples and the diversity of identities.
>
> We considered three configurations with 50%, 10%, and 1% of the available data for each subset, using two separate settings: (1) subsets sampled from the same set of identities (disjoint images), and (2) subsets sampled from disjoint identity sets. For each configuration, we projected a fixed set of opposite-gender features (e.g., Female) into both subspaces and computed the average cosine similarity between the two sets of projected vectors. We repeated the entire procedure in the reverse direction as well (i.e., projecting Male features into two independently constructed Female subspaces). These results are summarized in the following table:
>
> | Direction | Subset Setting | Data Ratio | Avg. Cosine Similarity |
> |--|--|--|--|
> | Female → Male | Same identities | 50% | 0.9959 |
> | Female → Male | Same identities | 10% | 0.9846 |
> | Female → Male | Same identities | 1% | 0.9023 |
> | Female → Male | Disjoint identities | 50% | 0.8839 |
> | Female → Male | Disjoint identities | 10% | 0.6936 |
> | Female → Male | Disjoint identities | 1% | 0.5280 |
> | Male → Female | Same identities | 50% | 0.9970 |
> | Male → Female | Same identities | 10% | 0.9843 |
> | Male → Female | Same identities | 1% | 0.9086 |
> | Male → Female  | Disjoint identities | 50% | 0.8866 |
> | Male → Female  | Disjoint identities | 10% | 0.7241 |
> | Male → Female  | Disjoint identities | 1% | 0.6061 |
>
> **Table C: Cosine Similarity Between Projected Features From Disjoint, but Same Attributed Subspaces**
>
> These results demonstrate that our subspace construction remains semantically robust even under considerable appearance and identity variation. Although a reduction in identity diversity (especially in the disjoint identity setting) leads to a moderate drop in projection consistency, the similarities remain relatively high overall than acceptance threshold 0.2432. This suggests that the use of diverse and sufficiently large identity pools enables the formation of robust attribute-specific subspaces.
>
> Furthermore, above results imply that increasing the number of identities is more crucial than the raw number of images when building robust subspaces. Therefore, curating and expanding well-annotated datasets with broad identity coverage for each attribute could be a promising direction for future work—particularly in enhancing both attack effectiveness and semantic control in adversarial face generation.
>
> ### **Response to Question 3 and 4 (Shortcoming of GAN-based Methods and Effectiveness of Diffusion Models):**
>
> Thank you for your thoughtful question. In the *Related Work* section, we discussed prior adversarial generation approaches that utilize GANs \[76, 24, 63, 41], as well as more recent efforts incorporating diffusion models, such as \[58]. While \[58] was not explicitly mentioned in the *Introduction*, we clarify here that it was selected as a representative comparison point for iterative solver-based methods operating in a white-box setting and relying on transferability. As one of the most recent works in this line of research (CVPR 2025), it serves as a comparison baseline in our evaluation. We include its performance in Tables 4 (attack success rates) and 5 (qualitative examples) of the main paper.
>
> That said, our critique of these methods—whether based on GANs or diffusion—is not aimed at the generative models themselves, but rather at their dependence on *gradient-based optimization*. This reliance necessitates surrogate models in black-box settings, inevitably pushing them toward the *transferability paradigm*, which often results in a significant drop in attack success rates due to distributional mismatches between the surrogate and target models. In contrast, as shown in Table 4, our method directly exploits the target model's information via score queries, offering more reliable black-box attack performance without relying on transferability.
>
> On the other hand, we would like to clarify that Arc2Face [54] is a diffusion-based model and one of the inverse models employed in our work. Thus, our approach already incorporates diffusion models within the adversarial generation pipeline. Specifically, Arc2Face serves as a high-quality inverse model that reconstructs pixel-level images from feature vectors, enabling us to generate realistic adversarial faces from our attribute-controlled projections. Furthermore, our framework also supports GAN-based inverse models such as Vec2Face [c], in a similar manner to NbNet [50], which is a non-diffusion-based inverse model based on deconvolutional networks. Although we could not evaluate Vec2Face due to the lack of publicly available code, we have already demonstrated the effectiveness of our method with NbNet. This underscores the generality of our approach across diverse inverse architectures.
>
> [b] Wright, John, et al. "Robust face recognition via sparse representation." IEEE transactions on pattern analysis and machine intelligence 31.2 (2008): 210-227.
>
> [c] Duong, Chi Nhan, et al. "Vec2face: Unveil human faces from their blackbox features in face recognition." Proceedings of the IEEE/CVF Conference on Computer Vision and Pattern Recognition. 2020.

---

> > ### Comment · Reviewer_phZN · 2025-08-03
> > **Good rebuttal**
> >
> > I read the response to my comments. I will keep my original score.

---

### Official Review · Reviewer_osyU · 2025-07-02

**Clarity:** 4
**Significance:** 3
**Originality:** 3
**Rating:** 4
**Confidence:** 3

**Summary:**

This paper proposes a novel non-adaptive, non-iterative black-box attack against face recognition systems, which constructs adversarial faces using a single batch query. The key insight is to exploit the geometric structure of deep face embeddings by identifying attribute-specific subspheres in the feature space. By projecting onto these subspheres and reconstructing faces through inverse models, the attack achieves high success rates against commercial APIs with minimal queries.

**Questions:**

-	The proposed subspace structure appears to be an empirically justified but heuristic assumption. Is there a more principled or data-driven way to discover alternative or more precise subspace structures that could potentially enhance attack effectiveness?
-	How well does the proposed method generalize to other biometric modalities such as gait or iris recognition? The paper focuses on face recognition, but the core idea might extend further if the subspace structure exists similarly in other domains.

**Ethical Concerns:**

["NO or VERY MINOR ethics concerns only"]

**Final Justification:**

I appreciate the authors' response. After considering their rebuttal and the feedback from my fellow reviewers, I have decided to maintain my original score.

**Limitations:**

Yes.

**Paper Formatting Concerns:**

N/A.

**Quality:**

3

**Strengths And Weaknesses:**

**Strengths**

-	Adversarial face generation is an important topic worthy of further investigation.
-	The non-iterative attack is efficient and practical for black-box FR systems.
-	This paper well connects theoretical supports and empirical validations.
-	Real-world demonstration convincingly highlights the societal impact of the proposed attack.

**Weaknesses**

-	More ablation studies on the projection quality across different inverse models could testify to the attack’s generality on inverse model choices.

---

> ### Author Rebuttal · Authors · 2025-07-28
>
> ### **Response to Weakness (Projection Quality across Different Inverse Models):**
>
> Thank you for pointing this out. Among the inverse models we experimented with, the well-trained ones—NbNet [50] and Arc2Face [54]—consistently generate high-quality reconstructions that accurately preserve the intended semantic attributes (see Figures 3(a), 7, and 11). In contrast, the inflexible and biased inverse model from [61], which relies on a fixed pre-trained StyleGAN, often fails to reflect the desired attributes, producing reconstructions that are systematically biased toward young white male faces (Figure 12). Notably, this issue is also acknowledged by the authors of [61].
>
> Precisely, all face feature vectors are extracted from the same recognition model, and each PCA-based projection matrix is computed using samples corresponding to a specific attribute (e.g., Male, Female, etc.). The resulting projected feature vectors—interpretable as adversarial features that remain close to the original identities but reflect different attributes—are obtained independently of the inverse model. Therefore, the *projection quality* primarily depends on how well the attributed subspheres are constructed, which in turn is influenced by the quality of attribute annotations in the attributed image dataset. This is evidenced by the successful reconstructions using Arc2Face and NbNet, where the reconstructed images consistently preserve the intended target attribute.
>
> However, when using the inverse model from \[61], we observe that reconstruction fails to reflect the intended attribute and is biased toward young white male faces. This issue stems from limitations in the inverse model itself, not from the projection quality. As such, although the projection step operates correctly, the final *reconstruction quality* is adversely affected by the inverse model's biases—potentially impacting the overall attack success rate.
>
> We will reflect this analysis more clearly in the final version of the paper and explicitly state that inverse model choice can affect the perceptual quality of generated adversarial faces, even when the projection is attribute-successful. Furthermore, we commit to including more qualitative examples of adversarial faces in the appendix—not only interpolations within principal components but also real-case adversarial samples per attribute, similar to Table 1.
>
>
> ### **Response to Question 1 (Subspace Structure – Heuristic vs. Principled Approach):**
>
> Thank you for this insightful question. While we report quantitative attack success rates only for gender and race attributes in the main paper, we also conducted the same principal component interpolation analysis for other attributes such as age, eyeglasses, baldness, and wearing a hat to qualitatively examine whether the semantic structure of the attribute-specific subspaces is preserved. As shown in Figures 9 and 10 (Appendix D), certain attributes like *baldness* exhibit consistent semantic variation along principal directions, suggesting the presence of a coherent subspace. In contrast, attributes like *wearing a hat* demonstrate only partial or inconsistent presence of the intended semantic attribute, indicating a weaker or noisier subspace.
>
>
> Several factors may contribute to this issue: noisy labels, insufficient sample size per attribute, or, in the worst case, the non-existence of a coherent attribute-specific subspace for certain attributes. Among these, we empirically demonstrate that increasing the number of attribute samples improves attack performance. In particular, we include results (see the following table and to be added in the final version) showing that using a larger set of attribute samples—e.g., constructing the gender-specific subspaces using CelebA [a] or VGGFace2 dataset instead of FairFace—leads to a consistent increase in attack success rates.
>
> | Attribute | Source Dataset | Number of Attributed Samples   | Attack Success Rate on $F_2$          | Attack Success Rate on $F_3$          |
> |:-----------:|:----------------:|:-------------:|:-------------------------:|-------------------------:|
> | Male      | FairFace       | 45,986      | 97.29% (Table 2)         | 98.28% (Table 23)        |
> | Female    | FairFace       | 40,758      | 93.33% (Table 2)         | 94.84% (Table 23)        |
> | Male      | CelebA        | 84,434       | 99.37% (+2.08)           | 98.48% (+0.20)           |
> | Female    | CelebA        | 118,165   | 99.53% (+6.20)           | 99.45% (+4.61)           |
> | Male      | VGGFace        | 1,958,913   | 99.48% (+2.19)           | 99.24% (+0.96)           |
> | Female    | VGGFace        | 1,349,127   | 99.77% (+6.44)           | 99.86% (+5.02)           |
>
> **Table A: Effect of Attribute Sample Size on Attack Success Rate (ASR)**
>
> To further investigate which attributes are more likely to yield coherent subspaces, we also conducted an auxiliary experiment (see our response to Reviewer VhZn – Question & Weakness 1) that compares the similarity between features projected onto subspaces constructed from samples with and without a given attribute. This analysis provides additional insights into the intrinsic structure of each attribute in the feature space. Nevertheless, we acknowledge that our current approach is not a fundamentally data-driven solution and still relies on manually defined attributes. To overcome this heuristic assumption, we believe that future work could explore unsupervised or weakly-supervised techniques—such as clustering methods (e.g., K-means)—to identify more intrinsic and potentially effective subspace structures for adversarial generation.
>
>
> ### **Response to Question 2 (Generalization to Other Biometric Modalities):**
>
> Thank you for the thoughtful question. As briefly mentioned in the introduction, we believe that extending our framework to other biometric modalities—such as voice—is a natural direction, especially for systems trained using metric learning approaches that enforce intra-class compactness and inter-class separability (e.g., feature vectors on a hypersphere). Such structural properties may allow the formation of attributed-specific subspaces in those domains as well.
>
> However, several challenges remain. As discussed in our response to weakness point, one key requirement for our method is the availability of a well-trained inverse model. While inverse models for face recognition are well-established—partly due to the standardized and cropped input format—learning reliable inverse models for modalities like voice is more difficult due to significant variations in input length and content. Similarly, gait recognition may face similar challenges due to the variable format of video sequences, which could make achieving high-quality reconstructions from projected feature vectors considerably harder, potentially hindering practical attack implementation.
>
> A second limitation concerns the availability and definition of meaningful attributes. As discussed in our response to Question 1, our framework relies on the ability to define attribute-specific subspaces based on semantically meaningful groupings such as gender or race. While such attributes are well-established and publicly annotated in the face domain, other modalities such as gait or iris may lack clearly defined, human-recognizable attributes that serve this purpose. Even when such attributes exist, collecting large-scale datasets with accurate annotations and sufficient intra-attribute variation remains a major obstacle. This issue further emphasizes the challenge of generalizing our method across modalities, where the subspace structure may be harder to define or less informative.
>
> Nonetheless, we believe that extending our method to other modalities is a compelling direction for future work, especially with the advancement of multimodal systems and generative (inverse) models across biometric domains.
>
> [a] Liu, Ziwei, et al. "Deep learning face attributes in the wild." Proceedings of the IEEE international conference on computer vision. 2015.

---

> ### Comment · Reviewer_osyU · 2025-08-04
>
> I appreciate the authors' response. After considering their detailed rebuttal and the feedback from my fellow reviewers, I have decided to maintain my original score. Although some ethics reviewers raised potential concerns that may warrant acknowledgment, I personally believe these issues are minor.

---

### Author Response · Authors · 2025-08-04
**Clarification on Real-World Testing and Dating Platform Mentions**

### **Clarifying the Scope and Ethics of Real-World Testing**

We acknowledge the reviewer’s concern regarding the broader societal implications of our real-world adversarial face images, especially when involving widely-used dating platforms. Therefore, in light of reviewer feedback and out of respect for potential concerns, we are fully willing to omit all mention of dating apps from the final manuscript. We clarify that the submitted version does not include any experimental results involving dating platforms. Instead, it merely stated that, upon acceptance and subject to the lifting of anonymization, we would release the results of dating-app verification experiments using only the authors’ own data, to illustrate the practical implications of the identified vulnerability. This mention was intended solely to demonstrate its real-world relevance, but it is not central to our main contribution. Our work fundamentally aims to reveal structural flaws in modern face recognition systems (FRSs) through rigorous analysis and controlled experimentation. The dating platform experiments were illustrative but not essential to validate the attack methodology or the proposed mitigation strategies.

While our study follows a long line of research targeting commercial face recognition APIs for vulnerability assessment [e, f, 37], we recognize that dating apps involve unique social sensitivities, such as user safety and trust, which demand an even higher level of ethical scrutiny. Unlike commercial APIs that serve primarily technical or enterprise-facing purposes, dating platforms operate in social contexts where misuse could have more immediate human consequences. Accordingly, we intend to retain only the commercial API evaluation (e.g., AWS), which aligns more closely with established norms in the security research community and poses minimal societal risk.

The following sections provide factual clarifications of the original setup, ethical guardrails, and technical motivations. Again, these are provided not as justification for keeping the dating platform contents, but to assure reviewers that even if such content were to remain, it was approached responsibly.

**Real-World Experiment Setup**

Our experiments used adversarial face images generated from self-captured author selfies as profile images on widely used dating platforms, with no third-party data involved. The verification process was completed using our authentic faces via video selfies, ensuring no interaction with other users. Immediately after testing whether adversarial images could pass verification, all accounts were deleted to eliminate any risk of harm or misuse. This setup minimized ethical concerns by restricting data to the authors’ own images and avoiding engagement with the platforms’ user bases, focusing solely on evaluating system robustness.

**On Terms of Service and Intent**

The Terms of Service (ToS) of major dating platforms prohibit “inauthentic” profiles but do not explicitly define acceptable profile images or address manipulated images derived from the user’s own selfies. Their verification processes—matching video selfies to profile images—tolerate significant variability, as legitimate users frequently fail due to factors like lighting, angles, or minor photo edits (e.g., filters). For instance, face authentication and liveness checks on mainstream dating platforms often tolerate some degree of mismatch between the profile photo and the video selfie due to factors like lighting, angle, or filters. As a result, the criteria for what constitutes a valid profile image remain somewhat ambiguous. Our experiments used adversarial profile images generated from our own selfies, paired with authentic video selfies, to test whether systems accept manipulated but self-derived images as plausible matches. This approach exploited a ToS gray area, as no explicit rules prohibit such images, and our tests complied with the technical verification process. Aligning with security research norms—where testing vulnerabilities without explicit ToS approval is standard when no harm occurs—we did not create fake profiles for interaction, but simply checked whether the system would interpret our generated samples as plausible matches for verification. This ensured ethical compliance while highlighting system weaknesses to improve FRS security.

---

### Author Response · Authors · 2025-08-04
**Ethics Statement, Checklist Revision, and Dual-Use Risk Mitigations**

**Checklist Misclassification and Ethics Statement**

We mistakenly selected [NA] for the human subjects checklist and fully acknowledge this error. At the time, we believed that using only self-generated selfies did not qualify as human subject research. We will revise this declaration accordingly in the final submission. Additionally, we will include a comprehensive ethics statement in the camera-ready version, detailing our use of self-data, the non-interactive nature of experiments, data privacy measures (e.g., immediate account deletion), and our commitment to responsible disclosure. Within 30 days of acceptance, we will formally contact the affected service providers (dating platforms), and AWS to share findings and support mitigation efforts, ensuring transparency and collaboration with stakeholders.

**Motivation and Publication Justification**

While our manuscript introduces an effective attack method, its primary aim is to provide a rigorous analysis that reveals structural weaknesses in face recognition systems, ultimately contributing to the development of more robust and trustworthy FRSs. Our work addresses a critical gap in FRS security by demonstrating how non-matching attributes (e.g., race, gender) can produce high cosine similarity scores due to flawed threshold optimization (typically 0.2–0.3). This vulnerability affects real-world applications like dating app verifications, risking unauthorized access or identity fraud. By exposing these flaws, our study informs service providers, regulators, and researchers about the need for robust FRS standards. For example, our findings could guide platforms to implement stricter verification thresholds or attribute-aware checks, enhancing user safety and trust. Section 5.3 proposes practical directions, such as integrating fairness-aware algorithms, to ensure equitable and secure systems. More concretely, we propose two defenses to mitigate the identified vulnerabilities:
1. Increasing feature vector dimensionality d and threshold tau to increase the number of queries k>=d*tau^2 for impersonation.
2. A new loss function (Appendix G.8) that maintains performance while enabling stricter thresholds.
These mitigations reduced attack success rates from 100% (e.g., VGG/White) to as low as 2.5% (e.g., VGG/Asian) across all tested scenarios, as detailed in Table 25 (Appendix). These results demonstrate the potential to secure FRSs against adversarial attacks under realistic threat models, offering actionable solutions for face recognition API service provider such as AWS.

| Target | Fair/Male | Fair/Female | VGG/White | VGG/Black | VGG/Asian |
|--------|-----------|-------------|-----------|-----------|-----------|
| $F_2$   | 98.53     | 96.69       | 99.94     | 99.49     | 98.30     |
| $F_3$  | 99.62     | 98.55       | 100.00    | 99.68     | 99.29     |
| $F_p$  | 6.02      | 3.07        | 15.52     | 4.55      | 2.50      |

**Table 25 (Appendix)**: $F_p$ denotes the prototype FRS, which shares the Inception ResNet-101 architecture with $F_3$ and is trained on the MS1MV3 dataset.

**Code Release and Dual-Use**

To prevent misuse, we will not release the adversarial generation pipeline or related APIs. FRS benchmarking code will be shared under controlled access (e.g., for academic researchers via a gated repository), ensuring responsible dissemination. Post-processing details critical to attack success will be disclosed only to service providers upon request, balancing transparency with risk mitigation.

**Demographic Targeting and Fairness**

Although our method can condition on attributes like gender or race, this was not intended for discriminatory targeting. Instead, we show that even with totally different attribute values, adversarial faces can be generated with high success—highlighting the structural vulnerabilities of the FRS. This motivates the need for future systems to integrate attribute-aware robustness.


**Conclusion**

Our work pioneers adversarial risk analysis in biometric FRSs, driving secure and fair systems through rigorous analysis and practical attack defense strategy. Furthermore, our safeguards—non-disclosure of post-processing steps, controlled code release, and collaboration with service providers—mitigate the potential misuse risks highlighted by the reviewers. We will expand ethical discussions in the camera-ready version, utilizing the additional page.


[e] Sharif, Mahmood, et al. "Accessorize to a crime: Real and stealthy attacks on state-of-the-art face recognition." Proceedings of the 2016 acm sigsac conference on computer and communications security. 2016.

[f] An, Shengwei, et al. "Mirror: Model inversion for deep learning network with high fidelity." Proceedings of the 29th Network and Distributed System Security Symposium. 2022.

---

### Decision · Program_Chairs · 2025-09-17

**Decision:**

Accept (poster)

**Comment:**

This paper presents a method for black-box generation of adversarial facial images that can impersonate a target identity in face recognition systems. Unlike prior work that relies on iterative optimization or surrogate models, the proposed approach exploits the geometric structure of deep face spaces. The authors observe empirically that similar attributes cluster into specific subspaces, which can be projected onto to efficiently generate adversarial images.

Strengths：

All reviewers agree that the paper is built on solid theoretical foundations and is supported by strong experimental validation. The paper is clearly written overall. Reviewer opinions are mixed on its novelty: for example, reviewer 2Myn finds the work presents novel insights, while reviewer phZN argues that the use of PCA is outdated. After evaluating the paper, the AC leans on the opinion of reviewer 2Myn: the novelty lies in the exploitation of feature subspaces for adversarial generation, while PCA is only a tool to realize the subspace.

Weaknesses：

There lies insufficiency in the analyses of related works. For example, reviewers osyU and phZN suggest to investigate alternative forms of subspace structures, reviewer 2Myn suggest to compare iterative-based SOTAs. In addition, all three ethical reviewer note ethical concerns, including sensitive data handling and societal impacts.

The most important reasons for the decision：

This paper presents novel insights on adversarial face generation which could benefit the community’s study and raise security awareness when employ FRS. The paper present overall solid contributions and is clear to read.

Summarization on the discussion and changes during the rebuttal period:

3 reviewers raise concerns regarding the analyses of related works. The authors respond to the concerns by supplementing discussions on subspace structures and some related works. After the rebuttal phase, reviewer 2Myn raised their score, and reviewers osyU and phZN keep their original justification as borderline accept. After reading the paper, the supplementary material as referred to by the authors, and the rebuttal discussion, the AC thinks the concerns are partly addressed, though some observations (e.g., the noisiness of subspace) is still partly empirical. All 3 ethical reviewers raised concerns and the authors respond through explicit ethical discussion. The AC thinks the responses overall makes sense yet the author should properly incorporate them into the manuscript through a further revision.

Overall comments:

This paper initially received 3 borderline accept reviews and 1 strong accept review. After the rebuttal phase, one reviewer raised their score to strong accept, resulting in unanimous acceptance. The paper investigates a non-adaptive adversarial attack against face recognition systems. All reviewers noted that the proposed methods are both theoretically and experimentally well grounded, and some further highlighted the paper’s novel insights. However, concerns remain regarding the need for comparisons with additional related works and alternative subspace structures, which are partly addressed during the rebuttal phase. In addition, 3 ethics reviewers expressed shared concerns about the use of human subjects and the broader societal impacts, which should be addressed through a further revision. Considering the overall novelty and quality of this work, the AC recommends acceptance as a poster, conditional on a satisfactory ethical revision.